# NAMPT-derived NAD$^+$ fuels PARP1 to promote skin inflammation through parthanatos cell death

**Francisco J. Martínez-Morcillo**[1,2], **Joaquín Cantón-Sandoval**[1,2], **Francisco J. Martínez-Navarro**[1,2], **Isabel Cabas**[1,2], **Idoya Martínez-Vicente**[2,3], **Joy Armistead**[4], **Julia Hatzold**[4], **Azucena López-Muñoz**[1,2], **Teresa Martínez-Menchón**[2,5], **Raúl Corbalán-Vélez**[2,5], **Jesús Lacal**[6], **Matthias Hammerschmidt**[4], **José C. García-Borrón**[2,3], **Alfonsa García-Ayala**[1,2], **María L. Cayuela**[2,5], **Ana B. Pérez-Oliva**[1,2]*, **Diana García-Moreno**[1,2]*, **Victoriano Mulero**[1,2]*

**1** Departamento de Biología Celular e Histología, Facultad de Biología, Universidad de Murcia, Spain, **2** Instituto Murciano de Investigación Biosanitaria-Arrixaca, Murcia, Spain, **3** Departamento de Bioquímica y Biología Molecular A e Inmunología, Facultad de Medicina, Universidad de Murcia, Murcia, Spain, **4** Institute of Zoology, Cologne Excellence Cluster on Cellular Stress Responses in Aging-Associated Diseases and Center for Molecular Medicine Cologne, University of Cologne, Cologne, Germany, **5** Hospital Clínico Universitario Virgen de la Arrixaca, Murcia, Spain, **6** Departamento de Microbiología y Genética, Facultad de Biología, Universidad de Salamanca, Spain

* anabpo@um.es (ABP-O); dianagm@um.es (DG-M); vmulero@um.es (VM)

**Data Availability Statement:** All relevant data are within the paper and its Supporting Information files.

## Abstract

Several studies have revealed a correlation between chronic inflammation and nicotinamide adenine dinucleotide (NAD$^+$) metabolism, but the precise mechanism involved is unknown. Here, we report that the genetic and pharmacological inhibition of nicotinamide phosphoribosyltransferase (Nampt), the rate-limiting enzyme in the salvage pathway of NAD$^+$ biosynthesis, reduced oxidative stress, inflammation, and keratinocyte DNA damage, hyperproliferation, and cell death in zebrafish models of chronic skin inflammation, while all these effects were reversed by NAD$^+$ supplementation. Similarly, genetic and pharmacological inhibition of poly(ADP-ribose) (PAR) polymerase 1 (Parp1), overexpression of PAR glycohydrolase, inhibition of apoptosis-inducing factor 1, inhibition of NADPH oxidases, and reactive oxygen species (ROS) scavenging all phenocopied the effects of Nampt inhibition. Pharmacological inhibition of NADPH oxidases/NAMPT/PARP/AIFM1 axis decreased the expression of pathology-associated genes in human organotypic 3D skin models of psoriasis. Consistently, an aberrant induction of NAMPT and PARP activity, together with AIFM1 nuclear translocation, was observed in lesional skin from psoriasis patients. In conclusion, hyperactivation of PARP1 in response to ROS-induced DNA damage, fueled by NAMPT-derived NAD$^+$, mediates skin inflammation through parthanatos cell death.

**Funding:** This work was supported by grants BIO2017-84702-R and PID2020-113660RB-I00 to VM and PhD fellowship to FJMN funded by MCIN/AEI/10.13039/501100011033 and European Regional Development Funds, grant 20793/PI/18 to VM funded by Fundación Séneca-Murcia, and contracts to ABPO, DGM and FJMM funded by Universidad de Murcia. The funders had no role in the study design, data collection and analysis, decision to publish, or preparation of the manuscript.

**Competing interests:** I have read the journal policy's and the authors of this manuscript have the following competing interests: A patent for the use of parthanatos inhibitors to treat psoriasis and atopic dermatitis has been registered by Universidad de Murcia and Instituto Murciano de Investigación Biosanitaria (#PCT/EP2020/083380).

**Abbreviations:** AIFM, apoptosis-inducing factor mitochondria associated 1; ANOVA, analysis of variance; ATRA, all-trans retinoic acid; CHT, caudal hematopoietic tissue; crRNA, crispr RNA; DAB, 3,3′-Diaminobenzidine; DMSO, dimethyl sulfoxide; dsBs, double-strand DNA break; FBS, fetal bovine serum; IDT, Integrated DNA Technologies; MeOH, methanol; MIF, migration inhibitory factor; NA, nicotinic acid; NAC, N-acetylcysteine; $NAD+$, nicotinamide adenine dinucleotide; NAM, nicotinamide; Nampt, nicotinamide phosphoribosyltransferase; NMN, nicotinamide mononucleotide; NMNAT, NMN adenylyltransferases; NP, N-phenylmaleimide; NR, nicotinamide riboside; PAR, poly(ADP-ribose); PARP, poly(ADP-ribose) polymerase; PBS, phosphate buffer saline; PFA, paraformaldehyde; ROI, region of interest; ROS, reactive oxygen species; RT, room temperature; Spint1a, serine protease inhibitor, kunitz-type, 1a; ssBs, single-strand DNA break; Th17, T helper 17; tracrRNA, trans-activating CRISPR RNA; TUNEL, terminal deoxynucleotidyl transferase dUTP nick end labeling; WIHC, whole-mount immunohistochemistry.

## Highlights

- Nicotinamide phosphoribosyltransferase (NAMPT) inhibition alleviates inflammation in zebrafish and human organotypic 3D skin models of psoriasis.

- NADPH oxidase–derived reactive oxygen species (ROS) mediate keratinocyte DNA damage and poly(ADP-ribose) polymerase 1 (PARP1) overactivation.

- Inhibition of parthanatos cell death phenocopies the effects of NAMPT inhibition in zebrafish and human psoriasis models.

- NAMPT and poly(ADP-ribose) (PAR) metabolism is altered in psoriasis patients.

## Introduction

Psoriasis is a noncontagious chronic inflammatory skin disease with global prevalence of 0.1% to 3% [1]. Despite being a relapsing and disabling disease that affects both physical and mental health, it is not usually life threatening. However, the cytokines and chemokines produced in the lesion may reach the blood and consequently cause comorbidities [2]. Although the etiology is still undetermined, external agents trigger inflammation in genetically predisposed epithelium [3,4]. Cytokines released by keratinocytes stimulate dendritic/Langerhans cells that drive specific T helper 17 (Th17) cell immune response and additional cytokines and chemokines close the inflammatory feedback loop that result in the skin lesion [5].

Nicotinamide adenine dinucleotide ($NAD^+$) is the most important hydrogen carrier in redox reactions in the cell, participating in vital cellular processes such as mitochondrial function and metabolism, immune response, inflammation, and DNA repair, among others [6]. $NAD^+$ levels are tightly regulated by Preiss–Handler, de novo, and salvage pathways [7]. Different tissues preferentially employ a distinct pathway regarding available precursors. The most important $NAD^+$ precursor is dietary niacin (also known as vitamin B3), consisting of nicotinamide (NAM), nicotinic acid (NA), and nicotinamide riboside (NR) [8]. NAM is also the product of $NAD^+$-consuming enzymes, that is why most mammalian tissues rely on NAM to maintain the $NAD^+$ pool via the $NAD^+$ salvage pathway [6,7]. The rate-limiting enzyme in the $NAD^+$ salvage pathway is nicotinamide phosphoribosyltransferase (NAMPT) that converts NAM into nicotinamide mononucleotide (NMN). After that, NMN adenylyltransferases (NMNAT 1–3) transform NMN into $NAD^+$ [8]. NAMPT has been associated with oxidative stress and inflammation [9], being identified as a universal biomarker of chronic inflammation, including psoriasis [10]. FK-866 is a noncompetitive highly specific NAMPT pharmacological inhibitor that induces a progressive $NAD^+$ depletion [11]. FK-866 has demonstrated anti-inflammatory effects in different experimental settings, including murine models of colitis and collagen-induced arthritis [12,13].

Several enzymes depend on $NAD^+$ to accomplish their biological functions. Poly(ADP-Ribose) (PAR) polymerases (PARPs) are major $NAD^+$-consuming enzymes that transfer ADP-ribose molecules (linear or branching PAR) to proteins or itself (auto-PARylation) [14]. PARPs are implicated in DNA repair and chromatin organization, gene transcription, inflammation and cell death, or stress responses, among others [7,14]. However, the main PARP biological function is to orchestrate the spatiotemporal repair of DNA damage; that is why

PARP1 is predominantly localized in the nucleus [7], being responsible for approximately 90% of PAR biosynthesis [14]. PARP1-3 are recruited and activated upon single- and double-strand DNA breaks (ssBs and dsBs) [14]. Several PARP inhibitors have been developed, such as olaparib, niraparib, rucaparib, talazoparib, and veliparib [15,16]. Once PARP1 is recruited to a ssB, the inhibitors prevent PARP enzymatic activity, entrapping and accumulating inactive PARP on DNA and triggering the collapse of replication forks, resulting in dsB generation during replication [15].

Under physiological conditions, DNA damage provoked by cellular metabolism is successfully handled by PARP1. However, alkylating DNA damage, oxidative stress, hypoxia, hypoglycemia, or activation of inflammatory pathways can trigger PARP1 hyperactivation. Excessive PARylation depletes cellular $NAD^+$ and ATP stores, although it does not directly imply cell death. However, the accumulation of PAR polymers and PARylated proteins reach the mitochondria causing depolarization of the membrane potential and apoptosis-inducing factor mitochondria associated 1 (AIFM1) release into the cytosol [17,18]. AIFM1 then recruits macrophage migration inhibitory factor (MIF) to the nucleus where AIFM1-MIF nuclease activity executes a large-scale DNA fragmentation resulting in a cell death pathway known as parthanatos [19].

In this work, we report a critical role played by $NAD^+$ and PAR metabolism in skin oxidative stress and inflammation. We found that hyperactivation of PARP1 in response to reactive oxygen species (ROS)-induced DNA damage, and fueled by NAMPT-derived $NAD^+$, mediates inflammation through parthanatos cell death in preclinical zebrafish and human organotypic 3D skin models of psoriasis. Consistent with this, clinical data support the alteration of $NAD^+$ and PAR metabolism in psoriasis, pointing to NAMPT, PARP1, and AIFM1 as novel therapeutic targets to treat skin inflammatory disorders.

## Results

### $NAD^+$ metabolites regulate skin oxidative stress and inflammation

In order to determine if $NAD^+$ metabolism has any role in the regulation of skin inflammation, we decided to perform functional experiments in the transgenic zebrafish line *lyz:dsRED*, which labels neutrophils. Manually dechorionated *lyz:dsRED* larvae were treated by bath immersion with different concentrations of $NAD^+$ from 24 hpf to 72 hpf (Fig 1A). Incubation with 1 mM $NAD^+$ resulted in a statistically significant increased neutrophil dispersion from the caudal hematopoietic tissue (CHT) compared to vehicle (DMSO) and 0.25 and 0.5 mM $NAD^+$-treated larvae (Figs 1A–1C and S1G). Despite the altered pattern of neutrophil distribution, some of which were present in the skin, both the integrity of the skin and its morphology were not affected (Fig 1C).

Given the role of $H_2O_2$ in driving neutrophil mobilization to acute [20] and chronic [21] insults, we used the $H_2O_2$ fluorescent probe acetyl-pentafluorobenzene sulphonyl fluorescein to know if this molecule was involved in the observed phenotype. $NAD^+$ treatment was able to enhance $H_2O_2$ production in skin in a dose-dependent manner compared to the control group (Figs 1D, 1E and S1G). Similar results were obtained with NAM (Fig 1F and 1G), a well-known $NAD^+$ booster [6], while NMN precursor was unable to increase skin oxidative stress (Fig 1F and 1G). Nevertheless, no differences in neutrophil redistribution were observed (Fig 1G). These effects are consistent with the recently reported ability of NAM supplementation to increase zebrafish larval $NAD^+$ but the failure of other $NAD^+$ precursors, including NMN [22]. In addition, it is already known that zebrafish larvae are able to take up $NAD^+$ and NADH [23–25], and this is not surprising since neural cells [25] and fibroblasts [26] are able to take up $NAD^+$ through connexin 43.

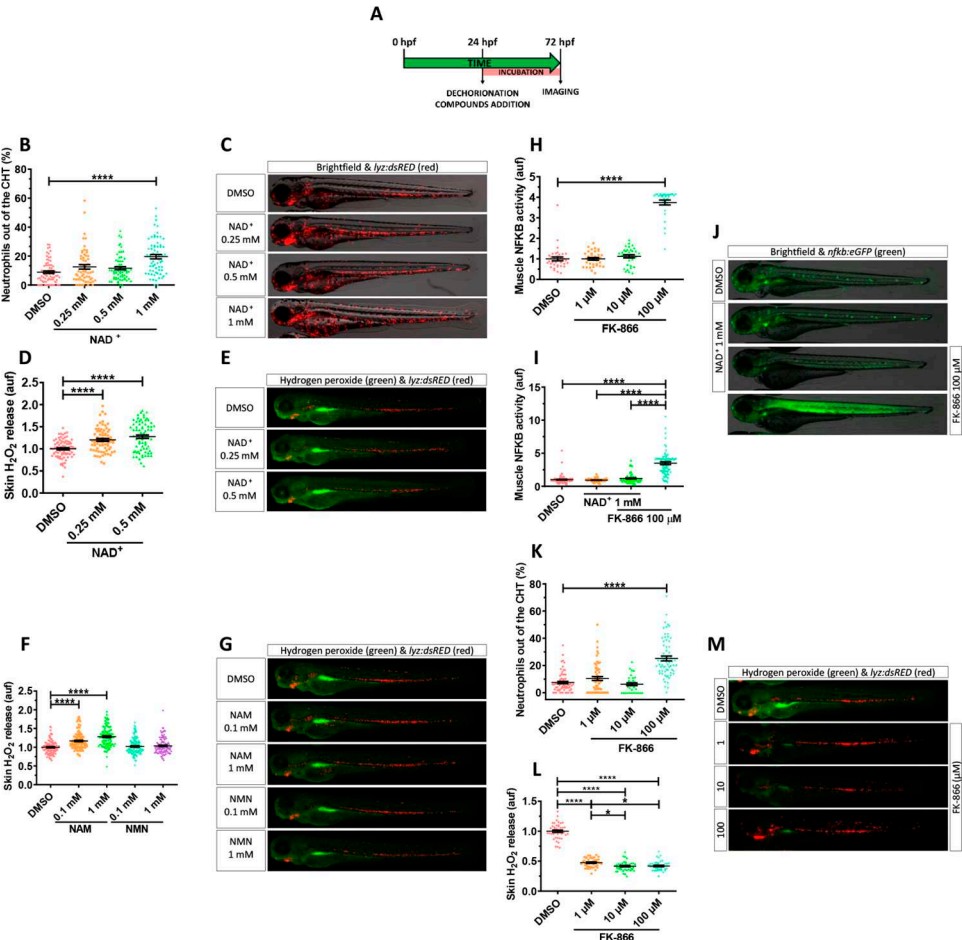

**Fig 1. NAD$^+$ metabolites regulate skin oxidative stress and inflammation. (A)** Embryos of 24 hpf were manually dechorionated, treated for 48 hours with NAD$^+$ metabolites or FK-866 by bath immersion, and images obtained at 72 hpf. **(B–M)** Quantification of the percentage of neutrophils out of the CHT in embryos treated with NAD$^+$ (0.25, 0.5 and 1 mM) (B). Representative merge images (brightfield and red channel) of *lyz:dsRED* zebrafish larvae of every group are shown (C). For H$_2$O$_2$ imaging, embryos were incubated in 50 μM of acetyl-pentafluorobenzene sulphonyl fluorescein solution for 1 hour. Quantification of fluorescence intensity for NAD$^+$-mediated (D) and NAM-/NMN-mediated (F) induction of H$_2$O$_2$ in the zebrafish skin. Representative merge images (green and red channel) of *lyz:dsRED* zebrafish larvae of every group are shown (E, G). NFKB activity was determined by quantification of fluorescence intensity in embryos treated with increasing doses of FK-866 (1, 10, and 100 μM) (H). Additionally, the influence of NAD$^+$ in the presence or absence of 100 μM FK-866 was assayed (I). Representative images (brightfield and green channel) of *nfkb:eGFP* zebrafish larvae of every group are shown (J). Neutrophil distribution in zebrafish embryos of 3 dpf treated with FK-866 (K), quantification of skin H$_2$O$_2$ production (L), and representative merge images (green and red channel) of *lyz:dsRED* zebrafish larvae of every group are shown (M). Each dot represents 1 individual and the mean ± SEM for each group is shown. *p*-Values were calculated using 1-way ANOVA and Tukey multiple range test. $^*p \leq 0.05$, $^{****}p \leq 0.0001$. The data underlying this figure can be found in S1 Data. ANOVA, analysis of variance; CHT, caudal hematopoietic tissue; NAD+, nicotinamide adenine dinucleotide; NAM, nicotinamide; NMN, nicotinamide mononucleotide.

We next wondered if the depletion of cellular NAD$^+$ stores by the well-characterized NAMPT inhibitor FK-866 [11] could also have an impact on skin oxidative stress and inflammation using the H$_2$O$_2$ probe and the transgenic line *nfkb:eGFP*, which accurately reports the activity of the master inflammation transcription factor NFKB [27]. FK-866 promoted a robust induction of NFKB activity in the muscle at the highest concentration used (100 μM) compared to control group (Fig 1H and 1J). Despite the NAD$^+$ ability to induce skin oxidative stress when used at 1 mM, it was unable to activate NFKB transcriptional activity in any tissue.

Importantly, NAD$^+$ effectively restored muscle NFKB activity in FK-866–treated larvae (Figs 1I, 1J and S1G), confirming the specificity of the inhibitor.

The inflammatory effect of FK-866 in muscle was confirmed by the robust neutrophil infiltration (Fig 1K and 1M). Furthermore, H$_2$O$_2$ production by skin keratinocytes was almost abolished by 1 μM of FK-866 (Fig 1L and 1M). Collectively, these results suggest that not only NAD$^+$ metabolite levels regulate oxidative stress in the skin and neutrophil infiltration, but also that low levels of NAD$^+$ trigger muscle inflammation.

## Inhibition of Nampt alleviates oxidative stress and skin inflammation in a zebrafish model of psoriasis

The influence of NAD$^+$ metabolism on skin oxidative stress and inflammation in wild-type zebrafish encouraged us to study its effect on the zebrafish psoriasis model with an hypomorphic mutation of *spint1a* (allele *hi2217*), which encodes the serine protease inhibitor, kunitz-type, 1a. Spint1a-deficient larvae showed increased H$_2$O$_2$ release in the skin compared with their wild-type siblings (Fig 2A and 2B). As described above for wild-type larvae, pharmacological inhibition of Nampt with FK-866 robustly decreased in a dose-dependent manner H$_2$O$_2$ production by Spint1a-deficient skin keratinocytes (Fig 2A and 2B).

Spint1a-deficient larvae show neutrophil infiltration in the skin [28–30]. We observed that 40% of neutrophils were out of the CHT in the mutants compared to 10% in wild-type larvae (Fig 2C and 2D). Mutant larvae treated with 10 or 50 μM FK-866 displayed a strong reduction of neutrophil dispersion (Fig 2C and 2D). However, 100 μM FK-866 induced muscle neutrophil infiltration, as observed in wild-type animals (Fig 2C and 2D). Importantly, epithelial integrity and keratinocyte aggregate foci were almost completely rescued in Spint1a-deficient larvae treated with 10 μM FK-866 (Fig 2E and 2F) or with another Nampt inhibitor (GMX1778) (S1A and S1B Fig). These results were further confirmed by genetic inhibition of the 2 zebrafish Nampt paralogues, namely Nampta and Namptb (S1C–S1F Fig). The low efficiency achieved by CRISPR/Cas-9 might indicate the indispensable role of Nampt activity in zebrafish development, as it has been found in mice [31]. In addition, NAD$^+$ supplementation exerted a negative effect on Spint1a-deficient skin. Thus, NAD$^+$ aggravates skin morphology alterations and neutrophil infiltration compared with wild-type animals (Fig 2G and 2H). In addition, NAD$^+$ treatment neutralized the beneficial effects of FK-866 on the *spint1a* mutant, worsening skin alterations and neutrophil infiltration (Fig 2G and 2H). Consistently, the high NFKB transcriptional activity observed in the skin of Spint1a-deficient larvae was reduced by FK-866 (Fig 2I and 2J). As expected, FK-866 and NAD$^+$ supplementation decreased and increased, respectively, NAD$^+$&NADH levels, as determined by ELISA (Fig 2K). However, no statistically significant differences between NAD$^+$&NADH levels in wild-type and Spint1a-deficient larvae were found (Fig 2K). The effects of FK-866 and NAD$^+$ supplementation on larval NAD$^+$ content were confirmed by using an enzymatic assay (Fig 2L). Collectively, these results indicate that Spint1a-deficient animals were more susceptible to NAD$^+$ supplementation than their wild-type siblings and that the beneficial effects of FK-866 on the skin were mediated by reducing skin NAD$^+$ availability.

## NADPH oxidase–derived ROS promote skin inflammation in Spint1a-deficient larvae

The higher levels of ROS in the skin of Spint1a-deficient larvae, together with their drastic reduction by pharmacological inhibition of Nampt, led us to hypothesize that Nampt-derived NAD$^+$ was fueling NADPH oxidases. We therefore used N-acetylcysteine (NAC), which can scavenge ROS directly and replenish reduced glutathione levels [32,33]. Spint1a-deficient

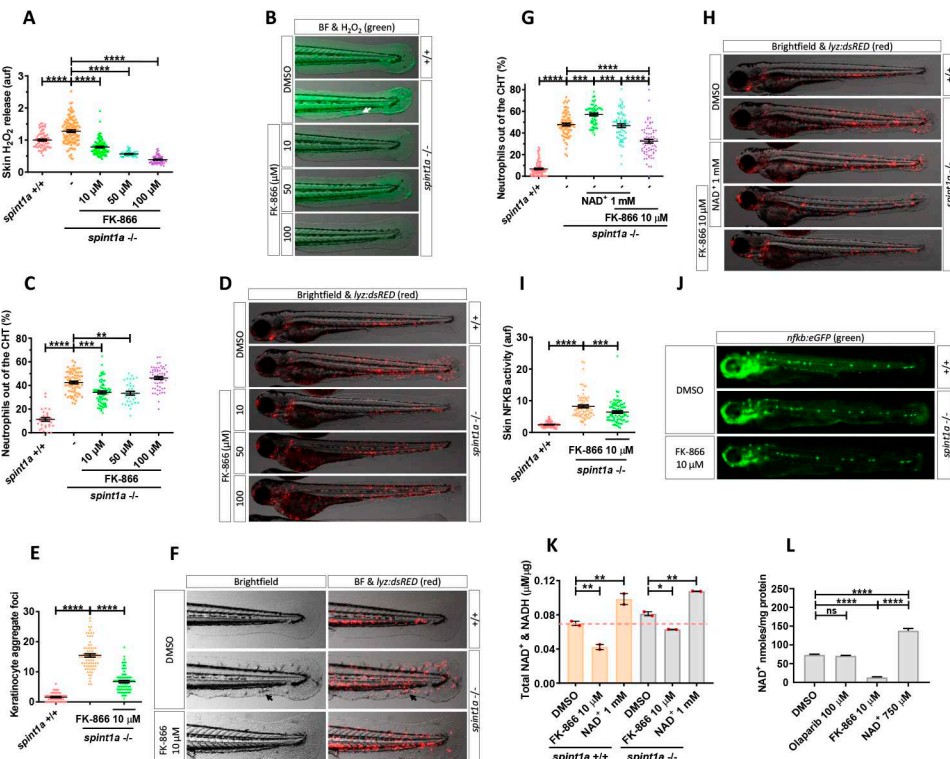

**Fig 2. Pharmacological inhibition of Nampt alleviates stress and skin inflammation in Spint1a-deficient larvae.**
**(A)** Quantification of skin $H_2O_2$ production in wild-type and Spint1a-deficient zebrafish larvae treated with FK-866 (0, 10, 50, and 100 µM). **(B)** Representative merge images (brightfield and green channel) of zebrafish larvae of every group are shown. The white arrow indicates a keratinocyte aggregate where $H_2O_2$ release was high. **(C)** Neutrophil distribution of wild-type and Spint1a-deficient zebrafish larvae treated with increasing doses of FK-866. **(D)** Representative merge images (brightfield and red channel) of *lyz:dsRED* zebrafish larvae of every group are shown. **(E)** Number of keratinocyte aggregates in the tail skin of larvae treated with 10 µM FK-866. **(F)** Detailed representative merge images (brightfield and red channel) of the tail of wild-type and Spint1a-deficient larvae treated with vehicle (DMSO) or 10 µM FK-866. Black arrows indicate keratinocyte aggregates and immune cell recruitment presents in nontreated Spint1a-deficient skin. **(G)** Neutrophil distribution of zebrafish embryos treated with 1 mM $NAD^+$ in the presence or absence of 10 µM FK-866. **(H)** Representative merge images (brightfield and red channels) of *lyz:dsRED* zebrafish larvae of every group are shown. **(I)** Quantification of fluorescence intensity of wild-type and Spint1a-deficient embryos treated with 10 µM FK-866. **(J)** Representative images (green channel) of *NF-kB:eGFP* zebrafish larvae of every group are shown. **(K)** Wild-type and Spint1a-deficient larvae of 72 hpf treated for 48 hours with 10 µM FK-866 and 1 mM $NAD^+$ were used for total $NAD^+$ and NADH determination by ELISA. (L) Spint1a-deficient larvae of 72 hpf treated for 48 hours with 100 µM olaparib, 10 µM FK-866 and 750 µM $NAD^+$ were used for $NAD^+$ determination by enzymatic cycling method. Each dot represents one individual, and the mean ± SEM for each group is also shown. *p*-Values were calculated using 1-way ANOVA and Tukey multiple range test. ns, not significant, $^{**}p \leq 0.01$, $^{***}p \leq 0.001$, $^{****}p \leq 0.0001$. The data underlying this figure can be found in S1 Data. ANOVA, analysis of variance; NAD+, nicotinamide adenine dinucleotide; Nampt, nicotinamide phosphoribosyltransferase.

larvae treated from 1 to 3 dpf with 100 µM NAC displayed a statistically significant reduction in skin neutrophil infiltration together with reduced skin alterations (Figs 3A, 3B, S2A and S2B). We next tested mito-Tempo, an antioxidant that specifically accumulates in the mitochondria imitating superoxide dismutase activity against superoxide and alkyl radical [34] and tempol, a nitroxide antioxidant that acts against the peroxynitrite decomposition compounds, nitrogen dioxide, and superoxide radical anion [35]. Whereas both antioxidants were able to reduce skin neutrophil infiltration (Fig 3C and 3D), NFKB activity (Fig 3E and 3F) and morphological alterations (S2C and S2D Fig) in Spint1a-deficient larvae, tempol was found to be much more potent than mito-Tempo (100 µM versus 100 nM). The relevance of cytosolic

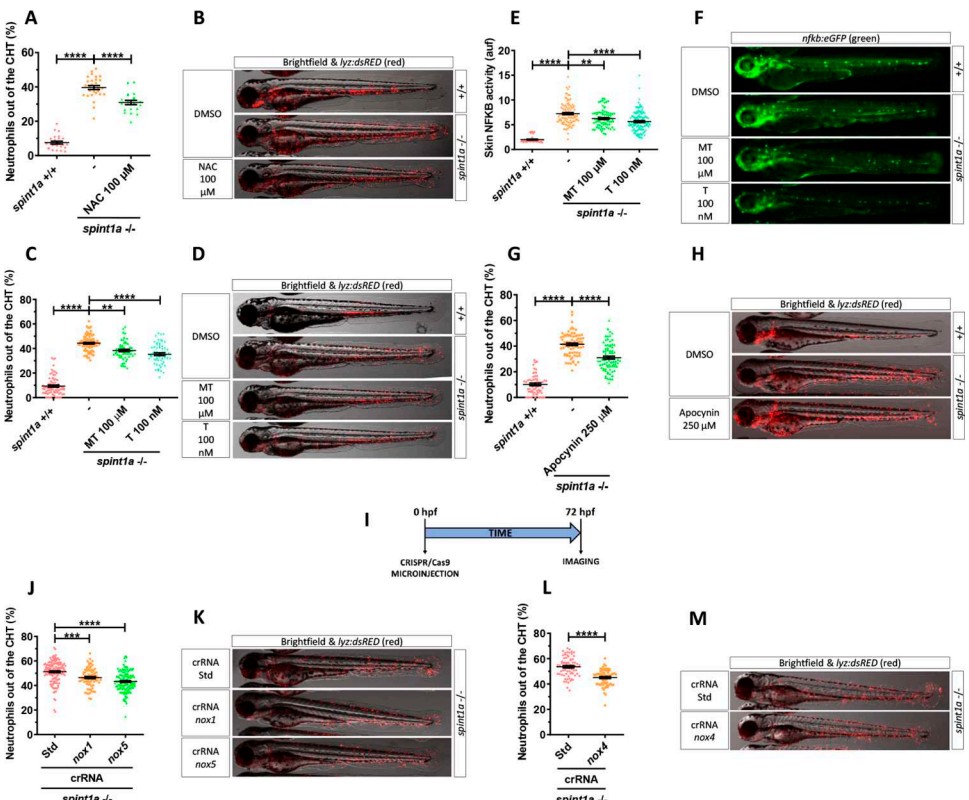

**Fig 3. NADPH oxidase–derived ROS promote skin inflammation in Spint1a-deficient larvae.** Quantification of neutrophil dispersion out of CHT of wild-type and Spint1a-deficient embryos treated with 100 μM NAC (**A**), 100 μM mito-TEMPO (MT) and 100 nM tempol (T) (**C**), 250 μM apocynin (**G**), or upon genetic inactivation of *nox1* and *nox5* (**I, J**) and nox4 (**I, M**) with CRISPR/Cas-9. Representative merge images (brightfield and red channels) of *lyz:dsRED* zebrafish larvae of every group are shown (**B, D, F, H, K, M**). Determination of NFKB transcriptional activity in the skin of embryos treated with MT and T (**E**) and representative images (green channel) of *nfkb:eGFP* zebrafish larvae of every group (**F**). Each dot represents one individual, and the mean ± SEM for each group is also shown. *p*-Values were calculated using 1-way ANOVA and Tukey multiple range test and *t* test. ns, not significant, ** $p \leq 0.01$, **** $p \leq 0.0001$. The data underlying this figure can be found in S1 Data. ANOVA, analysis of variance; CHT, caudal hematopoietic tissue; NAC, N-acetylcysteine; ROS, reactive oxygen species.

rather than mitochondrial ROS was further confirmed by the ability of pharmacological inhibition of NAPDH oxidases with apocynin to alleviate skin neutrophil infiltration of Spint1a-deficient larvae (Fig 3G and 3H) and skin alterations (S2E and S2F Fig). In addition, genetic inhibition of Nox1, Nox4, and Nox5 showed that all contributed to ROS production and skin inflammation (Figs 3I–3M, S2G–S2J and S3). Collectively, these results demonstrate that NADPH oxidase–derived ROS promote skin inflammation in Spint1a-deficient larvae.

## Inhibition of Parp1 alleviates oxidative stress and skin inflammation of Spint1a-deficient larvae

NAD$^+$ participates in more than 500 enzymatic reactions and regulates several key cellular processes [6]. We hypothesized that PARPs, which are major NAD$^+$-consuming enzyme [7], could be involved in skin inflammation. Pharmacological inhibition of Parps with olaparib efficiently rescued skin neutrophil infiltration (Fig 4A and 4B), NFKB activation (Fig 4C and 4D), and morphological alterations (S4A and S4B Fig) in Spint1a-deficient larvae. However, olaparib treatment failed to alter larval NAD+ content (Fig 2L). Consistently, genetic

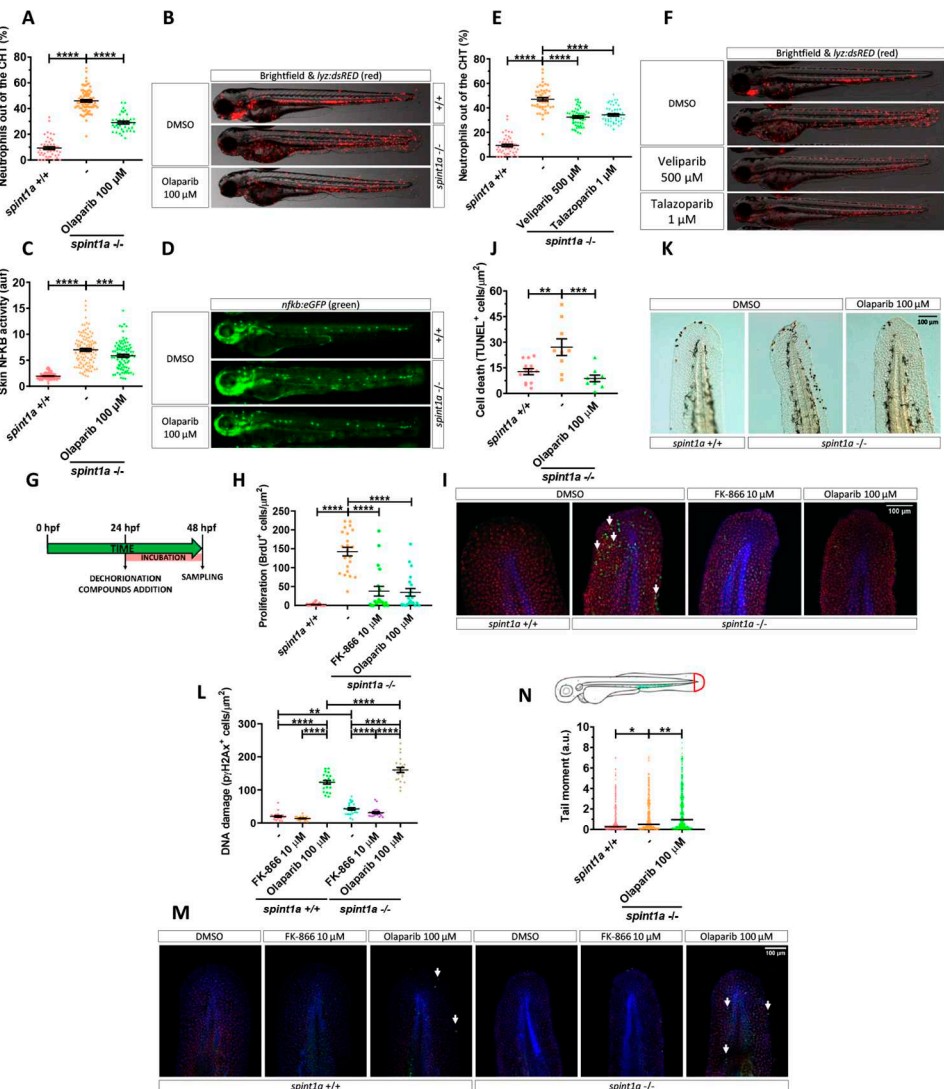

**Fig 4. Pharmacological inhibition of Nampt and Parp1 alleviates skin oxidative stress and inflammation and keratinocyte cell death, hyperproliferation and DNA damage in Spint1a-deficient larvae.** Analysis of neutrophil distribution (**A, E**) and NFKB transcriptional activity in the skin (**C**) of wild-type and Spint1a-deficient larvae treated with olaparib (**A, C**), and veliparib or talazoparib (**E**). Representative images (brightfield and red channel in B, F; green channel in D) of *lyz:dsRED* and *nfkb:eGFP* zebrafish larvae of every group are shown. Determination of BrdU positive cells from 48 hpf wild-type and Spint1a-deficient zebrafish larvae treated for 24 hours with 10 µM FK-866 or 100 µM olaparib (**G, H**). Representative merge images maximum intensity projection of a confocal Z stack from zebrafish larvae of every group are shown (**I**). WIHC with anti-BrdU (green, arrows), anti-p63 (red, basal keratinocyte marker) were counterstained with DAPI (blue). Quantification of TUNEL positive cells from 48 hpf wild-type and Spint1a-deficient zebrafish larvae treated for 24 hours with 100 µM olaparib (**J**). Representative images of zebrafish larvae of every group are shown (**K**). Quantification of pγH2Ax positive cells from 48 hpf wild-type and Spint1a-deficient zebrafish larvae treated for 24 hours with 10 µM FK-866 or 100 µM olaparib (**L**). Similarly, around 60 zebrafish tail folds (red boxed area) were amputated and disaggregated into cells for comet assay analysis in alkaline conditions (**N**). Representative merge images of maximum intensity projection of an apotome Z stack from zebrafish larvae of every group are shown (**M**). WIHC with anti-pγH2Ax (green, arrows), anti-P63 (basal keratinocyte marker, red) were counterstained with DAPI (blue). Scale bars, 100 µm. Each dot represents one individual. The mean ± SEM (A-L) and median (N) for each group is shown. *p*-Values were calculated using 1-way ANOVA and Tukey multiple range test (A-L) and Kruskal–Wallis test and Dunn multiple comparisons test (N). *$p \leq 0.05$, **$p \leq 0.01$, ***$p \leq 0.001$, ****$p \leq 0.0001$. The data underlying this figure can be found in S1 Data. ANOVA, analysis of variance; Nampt, nicotinamide phosphoribosyltransferase; Parp1, Poly(ADP-Ribose) polymerase 1; TUNEL, terminal deoxynucleotidyl transferase dUTP nick end labeling.

inhibition of Parp1 (S4C–S4F Fig) and treatment of larvae with other Parp inhibitors (veliparib and talazoparib) (Fig 4E and 4F) gave similar results, although the inhibitors showed different potencies (500 μM veliparib > 100 μM olaparib >1 μM talazoparib). Furthermore, increased PAR levels were found by western blot in the tail tissue of Spint1a-deficient larvae compared with their wild-type siblings (S4G Fig), which could be attenuated by both FK-866 and olaparib (S4H Fig). Similarly, Nampt (FK-866) and Parp (olaparib) inhibition significantly reduced skin epithelial lesions in the psoriasis mutant (S5A–S5C Fig), which share the skin inflammation and keratinocyte hyperproliferation phenotypes with the Spint1a-deficient line but has a loss-of-function mutation in *atp1b1a*, which encodes the beta subunit of a Na+/K+-ATPase pump [36,37].

## Inhibition of Nampt and Parp1 dampens keratinocyte hyperproliferation and cell death of Spint1a-deficient larvae

As the Spint1a-deficient phenotype starts with basal keratinocyte aggregation, mesenchymal-like properties acquisition, and cell death, leading to uncontrolled proliferation [28,29], we next analyzed whether Nampt or Parp1 inhibition affected keratinocyte proliferation and/or cell death. As early as 24 hours of treatment, pharmacological inhibition of either Nampt or Parps robustly reduced keratinocyte hyperproliferation in Spint1a-deficient larvae, assayed as BrdU incorporation (Fig 4G–4I), confirming the ability of both Nampt and Parp inhibition to reduce keratinocyte aggregates. Unexpectedly, olaparib also reduced the high number of terminal deoxynucleotidyl transferase dUTP nick end labeling (TUNEL)$^+$ keratinocytes found in Spint1a-deficient larvae (Fig 4J and 4K), despite the fact that Parp1 inhibition is expected to lead to the accumulation of DNA lesions and eventually to cell death [38]. We, therefore, investigated DNA damage analyzing the presence of the phosphorylated histone variant H2AX (pγH2Ax), which label dsBs and is independent of PARP1 [38], and performing a comet assay, which under alkaline conditions can detect both ssBs and dsBs [39]. The results showed higher keratinocyte DNA damage, assayed by either pγH2Ax staining (Fig 4L and 4M) or by comet assay (Fig 4N), in Spint1a-deficient keratinocytes. In line with previous studies, inhibition of Parp activity with olaparib increased DNA damage in wild-type and Spint1a-deficient larvae (Fig 4L and 4M). Remarkably, olaparib-induced DNA lesions were higher in Spint1a-deficient larvae (Fig 4L and 4N), suggesting increased susceptibility to Parp inhibition. Another interesting result was the reduced pγH2Ax staining in Spint1a-deficient keratinocytes in response to FK-866 treatment (Fig 4L and 4M). Taken together, these results suggest that Spint1a-deficient keratinocytes accumulate DNA breaks and have increased susceptibility to DNA stressors.

## Inhibition of parthanatos rescues skin inflammation of Spint1a-deficient larvae

Although the initial characterization of the Spint1a-deficient line revealed high keratinocyte cell death, it was refractory to inhibition of caspase or proapoptotic factors, suggesting an unidentified programmed cell death pathway [28,29]. Similarly, we did not find active caspase-3$^+$ keratinocytes in 2 dpf Spint1a-deficient or wild-type larvae (Fig 5A–5C). In addition, FK-866 or olaparib treatments did not induce apoptosis (Fig 5A–5C).

The absence of apoptosis in Spint1a-deficient larvae, together with the ability of Parp inhibitors to robustly reduce keratinocyte cell death in this model, led us to hypothesize that Parp1 overactivation in response to extensive ROS-mediated DNA damage would promote parthanatos cell death. To test this hypothesis, we used N-phenylmaleimide (NP), a chemical inhibitor of Aifm1 translocation from the mitochondria to the nucleus [40]. Treatment of Spint1a-deficient larvae with 10 nM NP showed statistically significant reduced skin neutrophil

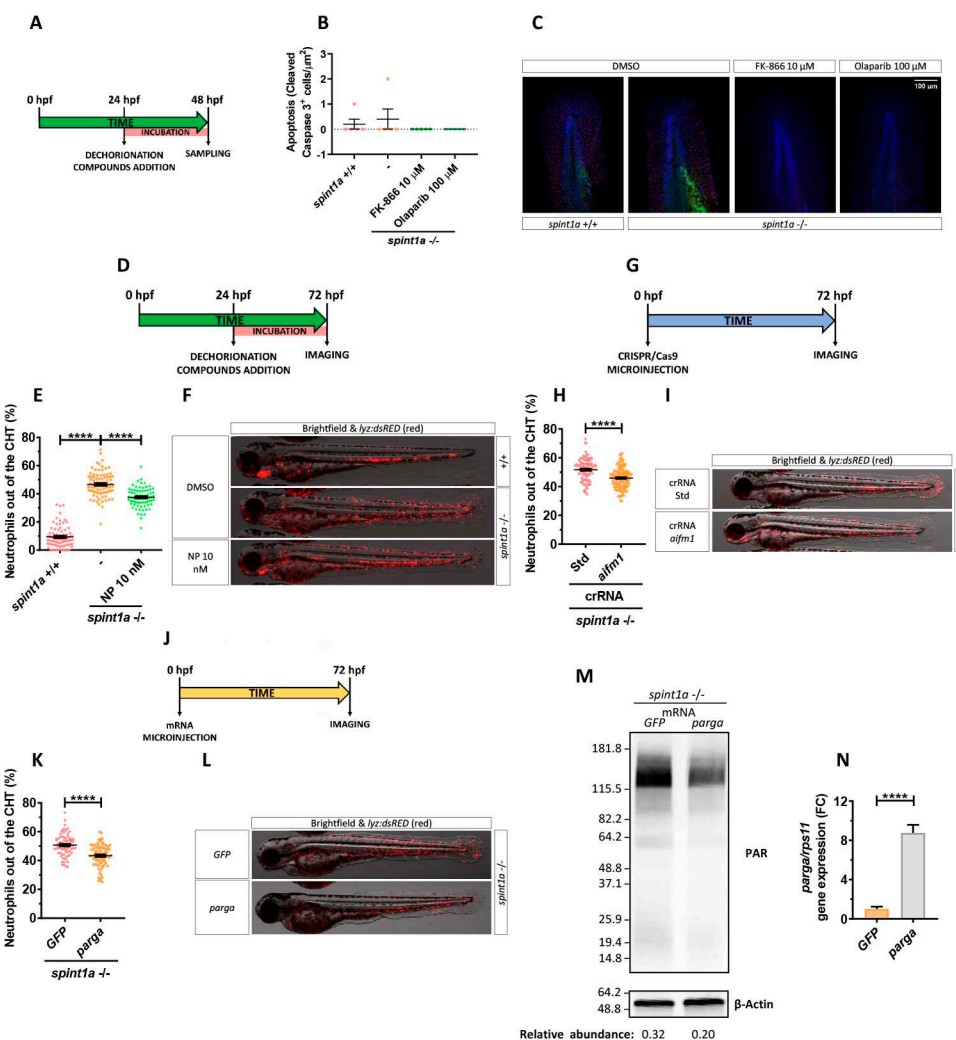

**Fig 5. Inhibition of parthanatos rescues skin inflammation of Spint1a-deficient larvae. (A, B)** Quantification of cleaved caspase 3 positive cells from 48 hpf wild-type and Spint1a-deficient larvae treated for 24 hours with 10 μM FK-866 or 100 μM olaparib. **(C)** Representative merge images of maximum intensity projection of an apotome Z stack from zebrafish larvae of every group are shown. WIHC with anti-cleaved Casp3 (green) and anti-P63 (basal keratinocyte marker, red) were counterstained with DAPI (blue). **(D–N)** Pharmacological and genetic inhibition experimental settings (D, G, J). Quantification of the percentage of neutrophils out of the CHT in embryos treated with 10 nM NP (Aifm1 translocation inhibitor) (E), *aifm1* genetic inhibition (H) and *parga* mRNA overexpression (K). Representative merge images (brightfield and red channels) of *lyz:dsRED* zebrafish larvae of every group are shown (F, I, L). For mRNA overexpression, 1-cell stage zebrafish eggs were microinjected, and imaging was performed in 3dpf larvae (J). Western blots with anti-PAR and anti-β-actin of tail fold lysates from 3 dpf wild-type and Spint1a-deficient zebrafish larvae microinjected with *parga* or *GFP* mRNA. The relative abundance of PAR with respect to β-actin is shown in each lane (M). Relative mRNA levels of 3 dpf wild-type zebrafish larvae microinjected with *parga* or *GFP* mRNA (N). Each dot represents one individual, and the mean ± SEM for each group is also shown. *p*-Values were calculated using 1-way ANOVA and Tukey multiple range test and *t* test. ****$p \leq 0.0001$. The data underlying this figure can be found in S1 Data. ANOVA, analysis of variance; CHT, caudal hematopoietic tissue; NP, N-phenylmaleimide.

infiltration (Fig 5D–5F) and keratinocyte aggregates (S6A and S6B Fig). Similar results were found upon genetic inhibition of Aifm1 (Figs 5G–5I and S5E–S5G) and forced expression of *parga*, which encodes PAR glycohydrolase a (Figs 5J–5N, S6H and S6I). Notably, a few kerati-nocytes localized in aggregates of Spint1a-deficient larvae showed nuclear Aifm1 staining and NP treatment blocked this translocation (S6C and S6D Fig). Collectively, these results further

confirm that overactivation of Parp1 in Spint1a-deficient animals mediates keratinocyte cell death through parthanatos rather than via depletion of ATP and NAD$^+$ cellular stores.

## Pharmacological inhibition of NADPH oxidases/NAMPT/PARP/AIFM1 axis decreased the expression of pathology-associated genes in human organotypic 3D skin models of psoriasis

The results obtained in zebrafish prompted us to study the impact of inhibition of oxidative stress, NAMPT and PARP1, in organotypic 3D human psoriasis models (Fig 6A). The results showed robust increase of NAMPT transcript levels in psoriatic epidermis, i.e., stimulated with cytokines IL17 and IL22 (Fig 6B). Pharmacological inhibition of NADPH oxidases with apocynin significantly reduced the mRNA levels of the inflammation marker defensin β4 (*DEFB4*) (Fig 6A), while it did not affect those of the differentiation markers filaggrin (*FLG*) and loricrin (*LOR*) (Fig 6C). Similarly, inhibition of NAMPT, PARP, and AIFM1 reduced the transcript levels *DEFB4* and *S100A8*, another inflammation marker associated with psoriasis (Fig 6B), while all treatments further reduced *LOR* and *FLR* mRNA levels, as did all-trans retinoic acid (ATRA) (Fig 6C). These results confirmed that inhibition of parthanatos cells death with NADPH oxidases, NAMPT, PARP, and AIFM1 inhibitors, also reduced inflammation in human psoriasis models.

## The expression of genes encoding NAD$^+$ and PAR metabolic enzymes is altered in psoriasis in humans

Transcriptomic analysis of human psoriasis lesional skin data collected in the GEO database revealed differential expression profile of genes encoding enzymes involved in NAD$^+$ and PAR

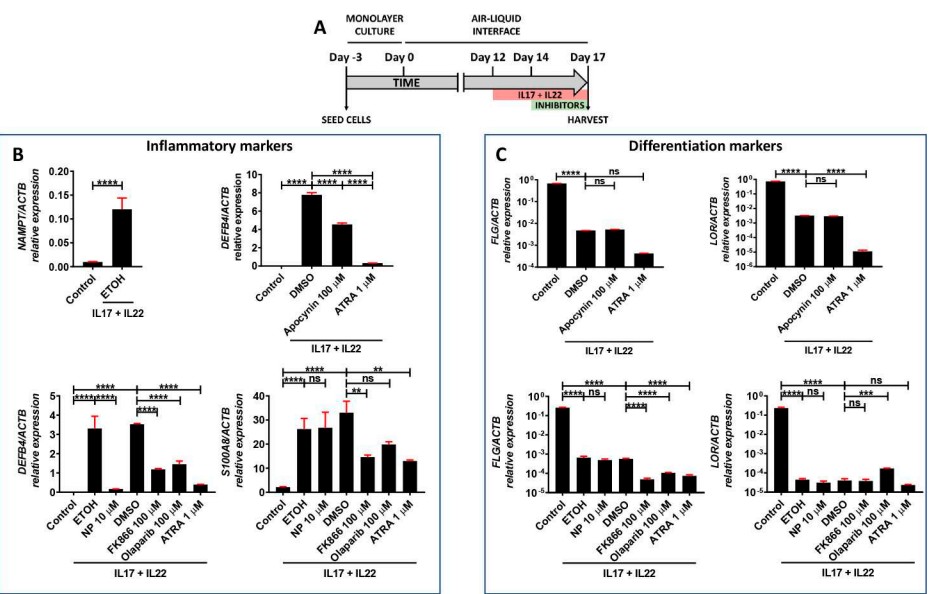

**Fig 6. Reduction of pathology-associated gene expression in human organotypic 3D skin models of psoriasis. (A–C)** Transcript levels of the indicated genes encoding inflammation (B) and differentiation (C) markers were determined in human organotypic 3D skin models pretreated with 30 ng/ml IL17A and IL22 in the presence of vehicles (ETOH and DMSO) or the indicated inhibitors. The mean ± SEM for each group is shown. The results are representative of 3 independent experiments. *p*-Values were calculated using 1-way ANOVA and Tukey multiple range test. ns, not significant. $^*p \leq 0.05$, $^{**}p \leq 0.01$, $^{***}p \leq 0.001$, $^{****}p \leq 0.0001$. The data underlying this figure can be found in S1 Data. ANOVA, analysis of variance; ATRA, all-trans retinoic acid; NP, N-phenylmaleimide.

(S7 and S8 Figs) metabolism. The mRNA levels *NAMPT* were upregulated (S8 Fig), while those of *NMNAT3* were down-regulated in psoriasis lesional skin (S8 Fig). The transcript levels of *NRK2*, which is involved NR conversion [41], were up-regulated in psoriasis lesional skin (S8 Fig). Preiss–Handler pathway seemed to show an intensified activity in psoriasis lesional skin, since up-regulation of the mRNA levels of genes encoding its components *NAPRT*, encoding nicotinate phosphoribosyltransferase, and *NADSYN*, encoding NAD synthetase, was observed (S8 Fig). Finally, the genes coding for NAD$^+$ biosynthetic enzymes involved in de novo pathway were also altered: While the transcript levels of *IDO1*, encoding indoleamine 2,3-dioxygenase 1, and *TDO2*, encoding tryptophan 2,3-dioxygenase, were induced in lesional skin, *QPRT*, encoding quinolinate phosphoribosyltransferase, slightly decreased in psoriasis compared to healthy skin (S7A Fig). With regard to genes encoding enzymes involved in NAD$^+$ degradation, increased *CD38* transcript levels were observed in psoriasis lesional skin (S8 Fig).

We next analyzed the transcript levels of *PARP1*, *AIFM1*, and *MIF*, which encode 3 indispensable parthanatos components [19] as well as those encoding different PAR hydrolases that negatively regulate protein PARylation. Transcriptomic data revealed strong increased mRNA levels of *PARP1*, *AIFM1*, and *MIF* in psoriasis lesional skin (S9 Fig). Although no alteration in the expression profile of the gene encoding *PARG* was found, the transcript levels of the genes encoding several PAR hydrolases, namely *MACROD1*, *MACROD2*, and *TARG1* were lower in psoriasis lesional skin (S9 Fig). Furthermore, genes encoding other PAR hydrolases were specifically up-regulated in psoriasis lesional skin (*ARH3*) or down-regulated (*NUDT16* and *ENPP1*) (S9 Fig). In addition, psoriasis lesional skin also showed enhanced transcript levels of *ARH1* (S9 Fig), whose product cleaves the terminal bond but only for targets PARylated on arginine [14]. These results taken together indicate that psoriasis may display increased PARylation.

## NAMPT and PAR increase and AIFM1 translocates to the nucleus in keratinocytes of human psoriasis lesions

Immunohistochemical analysis of samples from healthy skin and psoriasis lesions showed that NAMPT was hardly detected in healthy epidermis and dermis (Fig 7A). However, NAMPT was widely overexpressed in the spinous layer and in a few basal keratinocytes and dermal cells in psoriasis lesional skin (Fig 7A). Curiously, NAMPT immunoreactivity was mainly found in the nucleus of keratinocytes, but a fainter immunoreactivity was also observed in their cytoplasm (Fig 7A). Consistently, although PAR immunoreactivity was found in the nuclei of scattered keratinocytes of the spinous layer from healthy skin subjects, it was widely observed in the nuclei of most keratinocytes of the spinous layer and dermal fibroblasts from psoriasis lesional skin (Fig 7B), demonstrating increased PARylation in psoriasis lesional skin. Strikingly, a higher percentage of keratinocytes showing nuclear AIFM1 staining was found in lesional skin from psoriasis patients compared with healthy skin (Fig 7C and 7D).

## Discussion

NAD$^+$ metabolism plays a fundamental role in maintaining organism homeostasis. NAMPT, the rate-limiting step enzyme in the NAD$^+$ salvage pathway, has been associated with oxidative stress and inflammation [9], being identified as a universal biomarker of chronic inflammation, including psoriasis [10]. PARPs are major NAD$^+$-consuming enzymes involved in DNA repair, although their involvement in inflammation has also been widely recognized [7, 14]. We have shown here using the unique advantages of the zebrafish embryo/larval model for in vivo imaging and drug/genetic screening a crucial contribution of NAMPT and PARP1 to skin

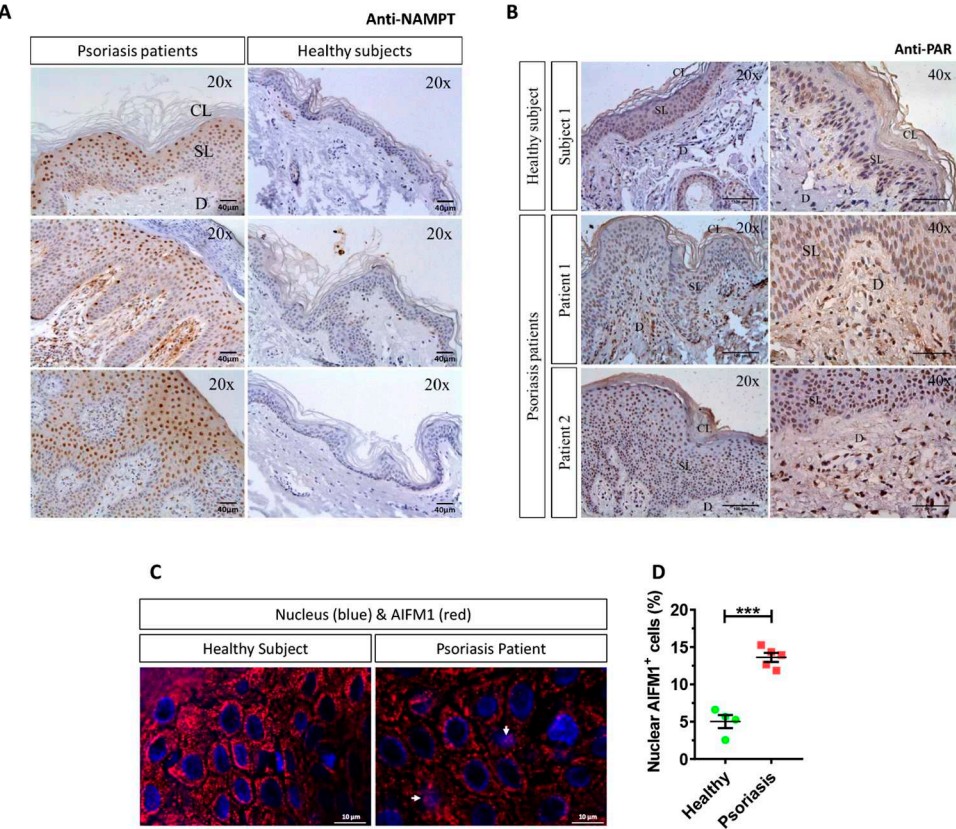

**Fig 7. Robust increase of NAMPT protein amount, PARylation, and nuclear translocation of AIFM1 in lesional skin from psoriasis patients. (A–C)** Representative images of sections from healthy and psoriatic skin biopsies that have been immunostained with an anti-NAMPT (A), anti-poly (ADP-ribose) (B), or anti-AIFM1 monoclonal antibodies and then slightly counterstained with hematoxilin (A, B) or DAPI (C). **(D)** The percentage of nuclear AIFM1 positive cells (white arrows) in human skin biopsies was calculated as the ratio between the number of keratinocytes in which AIFM1 is found in the nucleus and total keratinocyte number. Each point represents the mean of the quantification of 30 to 60 fields in section from an individual, and the mean ± SEM for each group is also shown. $p$-Values were calculated using $t$ test. ***$p \leq 0.001$. Scale bar is 40 μm in A, 100 μm and 50 μm in left and right panels of B and 10 μm in C. The data underlying this figure can be found in S1 Data. CL, cornified layer; D, dermis; Nampt, nicotinamide phosphoribosyltransferase; SL, spinous layer.

inflammation through the induction of parthanatos cell death (Fig 8). Consistently, inhibition of parthanatos also alleviated inflammation in human organotypic 3D models of psoriasis. In these models, the ability of $NAD^+$ and its precursors to induce oxidative stress can be explained by their capacity to boost NADH/NADPH intracellular levels that would fuel NADPH oxidases to generate ROS (Fig 8). In fact, pharmacological and genetic inhibition of NADPH oxidases or NAMPT efficiently counteracted skin $H_2O_2$ synthesis and inflammation in both zebrafish and human models. The relevance of oxidative stress in psoriasis has not been demonstrated definitively, but psoriasis patients show increased levels of oxidative stress markers, decreased levels of antioxidant molecules, and reduced activity of main antioxidant enzymes, such as superoxidase dismutase and catalase [42,43]. Additionally, high levels of oxidized guanine species, a marker of DNA/RNA damage, were found in the serum of psoriasis patients [44], further highlighting the relevance of the Spint1a-deficient line to study psoriasis. Curiously, high doses of FK-866 triggered NFKB and neutrophil infiltration into the muscle,

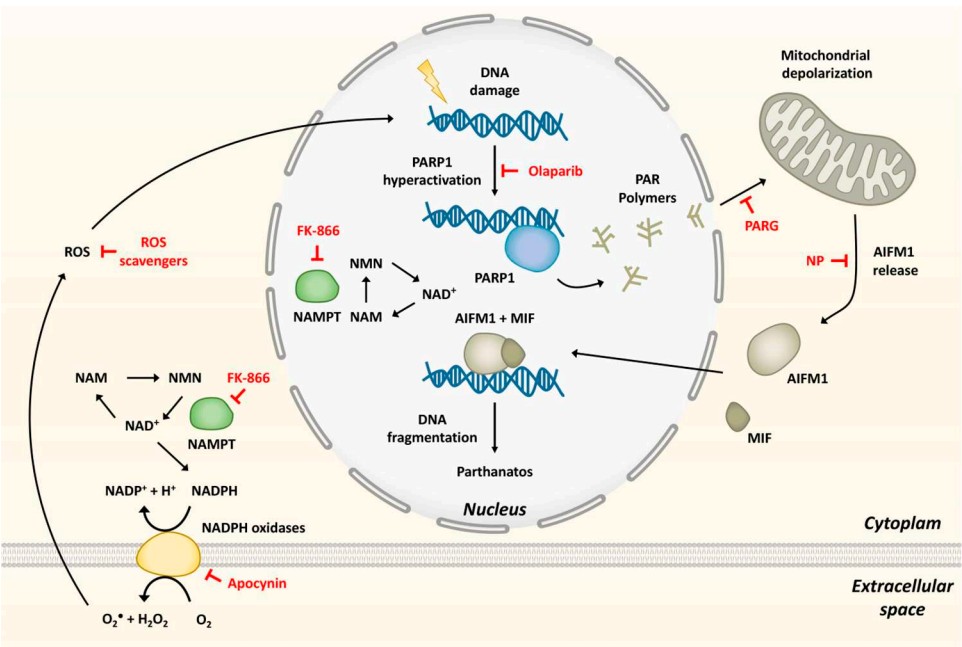

**Fig 8. Model showing that hyperactivation of PARP1 in response to ROS-induced DNA damage, and fueled by NAMPT-derived NAD+, mediates inflammation through parthanatos.** The inhibitors used are depicted in red. MIF, migration inhibitory factor; NAD+, nicotinamide adenine dinucleotide; NAM, nicotinamide; NAMPT, nicotinamide phosphoribosyltransferase; NMN, nicotinamide mononucleotide; NP, N-phenylmaleimide; Parp1, Poly(ADP-Ribose) polymerase 1; ROS, reactive oxygen species.

probably reflecting disruption of the cellular bioenergetics in this tissue due to critically low NAD$^+$ levels. This would need further investigation, but it is out of the scope of this study.

We also revealed new features of the Spint1a-deficient zebrafish line that may be useful in future studies. This model is of relevance for the study of human skin inflammatory disorders, since exacerbated serine protease activity is responsible for skin barrier defects and chronic inflammation in Netherton syndrome, a rare but severe autosomal recessive form of ichthyosis caused by mutation in *SPINK5*, which encodes the serine protease inhibitor LEKTI [45]. Notably, ablation of matriptase from LEKTI-deficient mice rescues skin barrier defects and inflammation [46]. Furthermore, mutations in *SPINK5* have also been associated with atopic dermatitis [47], further highlighting the relevance of the Spint1a-deficient zebrafish model. We found that this model exhibited increased H$_2$O$_2$ release in the skin, a well-known signal for leukocyte recruitment to acute [20] and chronic [21] insults. H$_2$O$_2$ was released at high levels in keratinocyte aggregates, precisely where higher NFKB levels were also observed and neutrophils were actively recruited. The relevance of oxidative stress in the skin inflammation of this line was confirmed by ROS scavenging and inhibition of NADPH oxidases and Nampt, being all able to restore skin morphology (Fig 8). We also found that Spint1a-deficient larvae exhibited extensive DNA damage, evaluated by pγH2Ax staining and comet assay and had increased susceptibility to olaparib, a DNA damage inducer [15]. This is an important observation from a clinical point of view, since psoriasis patients receiving PUVA have a significantly increased risk for the development of skin cancers (primarily squamous cell carcinoma) [48]. Pharmacological inhibition of Nampt not only reduced oxidative stress, skin inflammation, and keratinocyte hyperproliferation but also DNA damage. However, inhibition of Parp1 with olaparib increased keratinocyte DNA damage, despite being able to reduce at the same time

cell death and PARylation. Moreover, neither olaparib nor FK-866 treatments promoted keratinocyte apoptosis, contrasting previous studies where PARP inhibitors were reported to induce apoptosis in proliferating cells [38]. Therefore, olaparib induced an undesirable situation from a clinical point of view, where keratinocytes accumulated DNA damage but did not suffer apoptosis or another form of cell death. This is not surprising, since psoriatic keratinocytes are highly resistant to apoptosis [49], and its induction by PUVA may be involved in its therapeutic effects [50].

The depletion of NAD$^+$ cellular stores by Nampt inhibition might have also resulted in a direct inhibition of keratinocyte proliferation in the Spint1a-deficient and human organotypic 3D skin models of psoriasis. Indeed, it is known that cancer cells are more sensitive to the loss of NAMPT, a distinctive feature that was proposed to be used clinically in combination with other anticancer agents [9]. Moreover, various treatments for psoriasis target proliferation of keratinocytes and immune cells, such as acitretin, PUVA, methotrexate, and cyclosporine, among others [51]. However, no developmental alterations or delay were observed in larvae treated with Nampt or Parp inhibitors from 24 to 72 hpf. Therefore, although these inhibitors caused a blockage in cell proliferation, the results are consistent with a recovery of the mutant phenotype by the inhibition of oxidative stress and parthanatos.

One of the most interesting observations of this study is that Parp1 inhibition was able to restore epithelial homeostasis and to alleviate skin inflammation in both Spint1a- and Atp1b1a-deficient models. The ability of Aifm1 inhibition and Parga overexpression to phenocopy the effects of Nampt and Parp inhibition in the Spint1a-deficient model further confirmed that parthanatos induced by Parp1 overactivation, rather than by depletion of ATP and NAD$^+$ keratinocyte stores, mediated skin inflammation (Fig 8). This is consistent with the results obtained with Nampt inhibition, which resulted in depleted NAD$^+$ levels, but was also able to rescue skin inflammation and keratinocyte cell death and hyperproliferation in both zebrafish models. However, Nampt inhibition would simultaneously hamper NADPH oxidases and Parp1 enzymatic activities, while Parp inhibition would only suppress the latter. Anyway, prevention of cell death by either strategy might reduce danger signal and cytokine release, limiting immune cell recruitment and dampen inflammation [52] by blocking the inflammatory loop that propagates in psoriasis [5].

The analysis of human transcriptomic data from psoriasis lesional skin showed altered expression profiles of genes encoding NAD$^+$ metabolic enzymes, including salvage, Preiss–Handler and de novo pathways, and of genes encoding key enzymes involved in PAR metabolism and parthanatos. As psoriasis is characterized by an important cellular immune infiltration and growth of other cell types like nerves or blood vessels, the genes whose expression is altered in lesional skin may be expressed in different tissues other than keratinocytes. Our immunohistochemical analysis of psoriasis lesional skin confirmed the drastic induction of NAMPT at protein level and PAR accumulation in the nucleus of epidermal keratinocytes and dermal cells, together with the nuclear translocation of AIFM1, further confirming the relevance of the NAMPT/PARP1/AIFM1 axis in this disease.

In conclusion, we report that hyperactivation of Parp1 in response to ROS-induced DNA damage, and fueled by NAMPT-derived NAD$^+$, mediates inflammation through parthanatos cell death in preclinical zebrafish and human organotypic 3D skin models of psoriasis. The altered expression of genes encoding key enzymes involved in NAD$^+$ and PAR metabolism in psoriasis lesional skin, and in particular the robust induction of NAMPT, the accumulation of PAR and the nuclear translocation of AIFM1 in psoriasis lesional skin, point to NAMPT, PARP1, and AIFM1 as novel therapeutic targets to treat psoriasis and probably other skin inflammatory disorders.

## Methods

### Ethics statement

The experiments performed comply with the Guidelines of the European Union Council (Directive 2010/63/EU) and the Spanish RD 53/2013. Experiments and procedures were performed as approved by the Bioethical Committees of the University of Murcia (approval numbers #75/2014, #216/2014 and 395/2017) and Ethical Clinical Research Committee of The University Hospital Virgen de la Arrixaca (approval number #8/13).

### Animals

Wild-type zebrafish (*Danio rerio* H.) lines AB, TL, and WIK obtained from the Zebrafish International Resource Center (ZIRC) were used and handled according to the zebrafish handbook [53]. The transgenic zebrafish line *Tg(lyz:dsRED2)^{nz50}* [54] and Tg(NFκB-RE:eGFP) [27] were described previously. The mutant zebrafish lines *spint1a^{hi2217Tg/hi2217Tg}* [55] and the *atp-b1a^{m14/m14}* [37] were isolated from insertional and ehyl methanesulfonate induced mutagenesis screens, respectively.

### Genetic inhibition in zebrafish

The crispr RNA (crRNA) obtained from Integrated DNA Technologies (IDT) with the following target sequence were used: *aifm1* crRNA: 5′-CTTGCCAAGGTGGAGAACGG-3′, *parp1* crRNA: 5′-TGGATTTACTGACCTCCGCT-3′, *nampta* 5′-AGTAAAGAGCACATTTCC CCG-3′; *namptb* 5′-GGAGTAGACTTTATTTATAT-3′; *nox1* crRNA: CAAGCTGGTGGCC TACATGA; *nox4* crRNA: TTCGCTTGTGTCCTTCAAGC and *nox5* crRNA: GAGGTCATG GAAAATCTCAC. They were resuspended in duplex buffer at 100 μM, and 1 μl was incubated with 1 μl of 100 μM trans-activating CRISPR RNA (tracrRNA) at 95°C for 5 minutes and then 5 minutes at room temperature (RT) to form the complex. Then, the mix was diluted with 1.40 μl of duplex buffer. One μl of this complex was mixed with 0.20 μl of recombinant Cas-9 (10 mg/ml), 0.25 μl of phenol red, and 2.55 μl of duplex buffer. The efficiency of each crRNA was determined by the TIDE webtool (https://tide.nki.nl/) [56]. Crispant larvae of 3 dpf were used in all studies.

### Chemical treatments in zebrafish

Zebrafish embryos were manually dechorionated at 24 hpf. Larvae were treated from 24 hpf to 48 hpf or 72 hpf by chemical bath immersion at 28°C. Incubation was carried out in 6- or 24-well plates containing 20 to 25 larvae/well in egg water (including 60 μg/mL sea salts in distilled water) supplemented with 1% dimethyl sulfoxide (DMSO). The inhibitors and metabolites used and the concentrations tested and their targets are shown in S1 Table.

### Imaging of zebrafish larvae

Live imaging of 72 hpf larvae was obtained employing buffered tricaine (200 μg/mL) dissolved in egg water. Images were captured with an epifluorescence LEICA MZ16FA stereomicroscope set up with green and red fluorescent filters. All images were acquired with the integrated camera on the stereomicroscope and were analyzed to determine number of neutrophils and their distribution in the larvae. The transcriptional activity of NF-κB was visualized and measured with the zebrafish line NFκB-RE:eGFP.

$H_2O_2$ release was quantified employing the live cell fluorogenic substrate acetyl-pentafluorobenzene sulphonyl fluorescein (Cayman Chemical, Ann Arbor, Michigan, USA) [21,57]. Briefly, about 20 embryos of 72 hpf were rinsed with egg water and collected in a well of a

24-well plate with 50 μM of the substrate in 1% DMSO for 1 hour. ImageJ software was employed to determine mean intensity fluorescence of a common region of interest (ROI) placed in the dorsal fin for $H_2O_2$ production quantification. Similarly, a ROI located in muscle or skin was used to obtain mean intensity fluorescence of NFκB-RE:eGFP transgenic line.

## WIHC in zebrafish

BrdU incorporation assay was used to determine cell proliferation. Embryos of 48 hpf were incubated in 10 mM of BrdU dissolved in egg water for 3 hours at 28˚C followed by a 1-hour wash out with egg water and fixation in 4% paraformaldehyde (PFA) overnight at 4˚C or 2 hours at RT. For the rest of immunofluorescence techniques, embryos/larvae were directly fixed in 4% PFA, as indicated above. Embryos/larvae were then washed with phosphate buffer saline (PBS) with 0.1% tween-20 (PBST) 3 times for 5 minutes. In order to dehydrate the sample progressively, 25%/75% methanol (MeOH)/PBST, 50%/50% MeOH/PBST and 75%/25% MeOH/PBST and 100% MeOH were employed each for 5 minutes. At this point, embryos were stored at −20˚C. To proceed with the immunofluorescence, samples were re-hydrated in decreasing solutions of MeOH/PBST, as previously described, and then washed 3 times for 5 minutes with PBST. For BrdU staining next step consisted on applying 30 minutes a solution of 2N HCl in PBS supplemented with 0.5% Triton X-100 (PBSTriton), followed by 3 wash with PBSTriton for 15 minutes. Blocking step was carried out for at least 2 hours at RT with a PBSTriton solution supplemented with 10% fetal bovine serum (FBS) and 0.1% DMSO. The primary antibody incubation in blocking solution was done overnight at 4˚C or 3 to 4 hours at RT. After that, larvae were washed 6 times for 5 minutes. Incubation in secondary antibody in blocking solution was performed for 2 to 3 hours in the dark. In order to remove unbound secondary antibody, embryos were washed 3 times for 10 minutes with PBST. In this step, the sample was ready for DAPI staining with a DAPI solution (1:1,000) in PBST for 20 minutes followed by a wash out step of 3 times for 10 minutes with PBST. Finally, embryos were transferred to 80% glycerol/20% PBST and stored in the dark at 4˚C until imaging.

The following primary antibodies were used: rabbit anti-BrdU (Abcam (Cambridge, CB2 0AX, UK), ab152095, 1:200), mouse anti-p63 (Santa Cruz Biotechnology (69115 Heidelberg, Germany), sc-7255, 1:200), rabbit Anti-Active Caspase-3 (BD Bioscience (NJ 07417-1880, USA), #559565, 1:250), rabbit anti-H2AX.XS139ph (phospho Ser139) (GeneTex (CA 92606, USA), GTX127342, 1:200), and rabbit anti-AIFM1 (Merck (NJ 07033, USA), #AB16501, 1:200). Secondary antibodies were goat anti-rabbit Alexa Fluor 488 (#A-11008, 1:1,000) and Alexa Fluor 594 (#A-11012, 1:400) and goat anti-mouse Cyanine 3 (#A-10521, 1:1,000) (all from Thermo Fisher Scientific, MA 02451, USA). Images for BrdU staining were taken using a Zeiss Confocal (LSM710 META), while images for Aifm1 were acquired in a 1,024 × 1,024 pixel format in sequential scan mode between frames to avoid cross-talk, with objective used was HCX PL APO CS × 40, and the pinhole value was 1, corresponding to 114.73 μm. The other stains were acquired by ZEISS Apotome.2. All images were processed using ImageJ software.

## TUNEL assay in zebrafish

Embryos/larvae were fixed and dehydrated as described above. Afterward, embryos/larvae were rinsed with precooled (−20˚C) 100% acetone and then incubated in 100% acetone at −20˚C for 10 minutes. Samples were then washed 3 times for 10 minutes with PBST and incubated in a solution of 0.1% TritonX-100 and 0.1% sodium citrate (10%) in PBS for 15 minutes to further permeabilize the embryos/larvae. The next step consisted of rinsing specimens twice for 5 minutes in PBST. Following complete removal of PBST from samples, it was added 50 μL

of fresh TUNEL reaction mixture composed of 5 μL of enzyme solution mixed with 45 μL of labeling solution (In Situ Cell Death Detection kit, POD, Roche, Basel, Switzerland) for 1 hour at 37˚C, followed by 5 wash with PBST for 5 minutes. Blocking step was carried out for at least 1 hour at RT with blocking buffer. To proceed with TUNEL assay, blocking buffer was removed and added 50 μL Converter-POD (anti-fluorescein antibody conjugated to peroxidase) for 1 hour at RT or overnight at 4˚C on rocker. Embryos were rinsed 4 times for 30 minutes in PBST and incubated in 1 mL of 3,3′-Diaminobenzidine (DAB) solution for 30 minutes in the dark and transferred to a 24 well-plate. Two μL of a fresh 0.3% $H_2O_2$ solution was added to initiate peroxidase reaction that was monitored 10 to 20 minutes followed by rinsing and a wash out step of 2 times for 5 minutes with PBST. Finally, embryos were transferred to 80% glycerol/20% PBST and stored in dark at 4˚C until imaging. Images were acquired by ZEISS Apotome.2 and processed using ImageJ software.

## Comet assay in zebrafish

Zebrafish embryos at 48 hpf were anesthetized in tricaine (200 μg/mL) dissolved in egg water, and the end of the fin fold was amputated with a scalpel. Tissues collected from around 60 embryos were pooled and then spun and resuspended in 1 mL PBS. Liberase at 1:65/volume of PBS (Roche, cat # 05401119001) was added, and tissues were incubated at 28˚C for 35 minutes, pipetting up and down every 5 minutes. To stop the reaction, FBS was added to a final concentration of 5% in PBS. From now on, samples were kept on ice. Disaggregated fin folds were filtered through a 40 μM filter and washed using PBS + 5% FBS. Cell suspension was centrifuged at 650xg for 5 minutes and resuspended in 50 μL of PBS + 5% FBS. In order to determine cell number, Trypan Blue-treated cell suspension was applied to Neubauer chamber and cell were counted in an inverted microscope. Around 15,000 cells were employed to perform the Alkaline Comet Assay according to the manufacturer's protocol (Trevigen, MD 20877, USA). Briefly, cells were added in low melting point agarose at 37˚C at a ratio of 1:10 (v/v) and then were placed onto microscope slides. After adhesion at 4˚C for 30 minutes in the dark, slides were immersed in lysis buffer (precooled at 4˚C) overnight at 4˚C. Next, DNA was unwound in alkaline electrophoresis solution pH>13 (200 mM NaOH, 1 mM EDTA) at RT for 20 minutes in dark, followed by electrophoresis run in the same buffer at 25 V (adjusting the current to 300 mA) for 30 minutes. Slides were washed twice in distilled water for 5 minutes and in 70% ethanol for 5 minutes, and then they were dried at 37˚C for 30 minutes. Finally, DNA was stained with SYBR Green I Nucleic Acid Gel Stain 10,000X (Thermo Fisher Scientific, MA 02451, USA), and images were taken using a Nikon Eclipse TS2 microscope with 10× objective lens. Quantitative analysis of the tail moment (product of the tail length and percent tail DNA) was obtained using CASPLAB software. More than 100 randomly selected cells were quantified per sample. Values were represented as the median of the tail moment of treated cells relative to the median of the tail moment of untreated cells.

## Western blot

Zebrafish embryos at 72 hpf were anesthetized in tricaine (200 μg/mL) dissolved in egg water, and the end of the fin fold was amputated with a scalpel. Tissues collected from around 120 embryos were pooled and then spun and resuspended in 80 μL of 10 mM Tris pH 7.4 + 1% SDS. Samples were then incubated at 95˚C for 5 minutes with 1,400 rpm agitation, followed by maximum speed centrifugation for 5 minutes. Supernatants were frozen at −20˚C until proceeding. BCA kit was employed to quantify protein using BSA as a standard. Fin lysates (10 μg) in SDS sample buffer were subjected to electrophoresis on a polyacrylamide gel and transferred to PVDF membranes. The membranes were incubated for 1 hour 30 minutes with

Tris Buffered Saline, with Tween 20, pH 8.0 (TTBS) containing 5% (w/v) skimmed dry milk powder and immunoblotted in the same buffer 16 hours at 4°C with the mouse monoclonal antibody to human PAR (1/400, ALX-804-220, Enzo, Life Sciences, NY 10022, USA). The blot was then washed with TTBS and incubated for 1 hour at RT with secondary HRP-conjugated antibody diluted 2,500-fold in 5% (w/v) skimmed milk in TTBS. After repeated washes, the signal was detected with the enhanced chemiluminescence reagent and ChemiDoc XRS Bio-Rad.

## Total NAD$^+$ and NADH determination

Zebrafish embryos at 72 hpf were anesthetized in tricaine (200 μg/mL) dissolved in cold PBS in order to amputate the tail at the end of the yolk sac extension with a scalpel. Tissues from around 120 embryos were pooled and collected in lysis buffer provided by the kit (ab186032, Abcam) according to the manufacturer's protocol. Tissues were homogenized and centrifuged at 1,400 rpm for 5 minutes at 4°C. Supernatants were collected and centrifuged at maximum speed for 10 minutes at 4°C. Supernatants were employed to protein quantification with BCA kit using BSA as a standard. To proceed with Total NAD and NADH determination, 50 μg of protein were employed.

## NAD$^+$ determination

Around 40 embryos of 3dpf were deyolked using calcium-free zebrafish Ringer's solution (116 mM NaCl, 2.9 mM KCl, 5 mM HEPES, pH 7.2). The NAD$^+$ content of the samples was measured by using an enzymatic spectrophotometric cycling assay based on the coupled reaction of malate and alcohol dehydrogenases [58].

## Analysis of parga expression

Total RNA was extracted from whole 2 dpf embryos (60) with TRIzol reagent (Thermo Fisher Scientific) and treated with DNase I, amplification grade (1 U/mg RNA; Invitrogen). Super-Script VILO cDNA Synthesis Kit (Thermo Fisher Scientific) was used to synthesize first-strand cDNA from 1 mg of total RNA. Real-time PCR was performed with an ABI PRISM 7500 instrument (Thermo Fisher Scientific, MA 02451, USA) using Power SYBR Green Master Mix. Gene expression was normalized to the ribosomal protein S11 (rps11) content in each sample following the Pfaffl method [59]. The primers used are the following: 5′-CCACTGA GCCTGATAGCCAG-3′ and 5′-ATTCTCTATTGGGGTCCCGC-3′ for *parga* and 5′-GGC GTCAACGTGTCAGAGTA-3′ and 5′-GCCTCTTCTCAAAACGGTTG-3′ for *rsp11*. PCR was performed with triplicate samples and repeated with samples from 2 independent experiments.

## Gene Expression Omnibus (GEO) database

Human psoriasis (accession number: GSD4602) transcriptomic data were collected in the GEO database (https://www.ncbi.nlm.nih.gov/geo/). Gene expression plots were obtained using GraphPad Prism Software.

## Immunohistochemistry in human skin samples

Skin biopsies from healthy donors (*n* = 5) and psoriasis patients (*n* = 6) were fixed in 4% PFA, embedded in Paraplast Plus, and sectioned at a thickness of 5 μm. After being dewaxed and rehydrated, the sections were incubated in 10 mM citrate buffer (pH 6) at 95°C for 30 minutes and then at RT for 20 minutes to retrieve the antigen. Afterward, steps to block endogenous

peroxidase activity and nonspecific binding were performed. Then, sections were immunostained with mouse monoclonal antibodies to NAMPT (sc-166946, Santa Cruz Biotechnology, 1/100), a poly (ADP-ribose) (ALX-804-220; Enzo Life Sciences (NY 10022, USA), 1/100), or rabbit anti-AIFM1 (Merck, #AB16501, 1:200) followed by 1/100 dilution of biotinylated secondary antibody followed by ImmunoCruz goat ABC Staining System (sc-2023, Santa Cruz Biotechnology) or goat anti-rabbit Alexa Fluor 594 (Thermo Fisher Scientific) (#A-11012, 1:400). Finally, after DAB staining solution was added, sections were dehydrated, cleared, and mounted in Neo-Mount or directly stained with DAPI solution as indicated above. No staining was observed when primary antibody was omitted. Sections stained with DAB were finally examined under a Leica microscope equipped with a digital camera Leica DFC 280, and the photographs were processed with Leica QWin Pro software. Sections stained with fluorescence were examined under a Stellaris confocal microscope with a 63× objective and further processed with Leica software.

## Human organotypic 3D models

Insert transwells (Merck, MCHT12H48) were seeded with $10^5$ human foreskin keratinocytes (Ker-CT, ATCC CRL-4048) on the transwells in 300 μL CnT-PR medium (CellnTec) in a 12 well format. After 48 hours, cultures were switched to CnT-PR-3D medium (CELLnTEC, Bern 3014, Switzerland) for 24 hours and then cultured at the air–liquid interface for 17 days. From day 12 to 17 of the air–liquid interphase culture, the Th17 cytokines IL17A (30 ng/mL) and IL22 (30 ng/mL) were added [60]. Pharmacological treatments were applied from day 14 to 17 and consisted of 100 μM apocynin, 100 μM FK-866, 100 μM olaparib, 10 μM NP and 1 μM ATRA. Culture medium was refreshed every 2 days. At day 17, the tissues were harvested for gene expression analysis.

## Statistical analysis

Data were analyzed by analysis of variance (ANOVA) and a Tukey multiple range test to determine differences between groups with Gaussian data distribution (square root transformation were employed for percentage data). Nonparametric data were analyzed by Kruskal–Wallis test and Dunn multiple comparisons test. The differences between 2 samples were analyzed by the Student *t* test. The contingency graphs were analyzed by the chi-squared (and Fisher exact) test.

## Supporting information

**S1 Fig. Related to Fig 2. Genetic and pharmacological inhibition of Nampt alleviates skin inflammation and restores epithelial integrity in Spint1a-deficient larvae. (A)** Neutrophil distribution of wild-type and Spint1a-deficient larvae treated with the pharmacological inhibitors of Nampt GMX1778 and FK-866. **(B)** Representative merge images (brightfield and red channels) of *lyz:dsRED* zebrafish larvae of every group are shown. **(C)** For genetic inhibition using CRISPR/Cas-9 technology, 1-cell stage zebrafish eggs were microinjected and imaging was performed in 3 dpf larvae **(D)**. Quantification of the percentage of neutrophils out of the CHT in Spint1a-deficient larvae upon knockdown of Nampta/Namptb. **(E)** Representative merge images (brightfield and red channel) of *lyz:dsRED* zebrafish larvae of every group are shown **(F)** Analysis of genome editing efficiency in larvae injected with control or *nampta* and *namptb* crRNA/Cas-9 complexes and quantification rate of NHEJ-mediated repair showing all INDELs (https://tide.nki.nl/). Each dot represents one individual, and the mean ± SEM for each group is also shown. *p*-Values were calculated using 1-way ANOVA and Tukey multiple range test and *t* test. \*\*\**p* ≤ 0.001, \*\*\*\**p* ≤ 0.0001. **(G)** Inflammation and oxidative stress

determination in zebrafish larvae [21]. Inflammation was scored by using 2 different approaches: (i) the *lyz:dsRED* zebrafish transgenic line was used to quantify the percentage of neutrophils out of the CHT, i.e., neutrophil dispersion; and (ii) the *nfkb:eGFP* zebrafish transgenic line was used to determine NFKB activity by quantification of fluorescence intensity in the drawn white box. For oxidative stress, analysis of fluorescence intensity of the ROI (white box) of larvae preloaded with an $H_2O_2$ fluorogenic probe. The data underlying this figure can be found in S1 Data. ANOVA, analysis of variance; CHT, caudal hematopoietic tissue; INDEL, insertion and deletion; Nampt, nicotinamide phosphoribosyltransferase; NHEJ, nonhomologous end joining; ROI, region of interest.
(PDF)

**S2 Fig. Related to Fig 3. ROS scavenging and inhibition of NADPH oxidases rescue skin neutrophil recruitment and skin morphological alterations of Spint1a-deficient larvae.** Quantification of keratinocyte aggregation foci in the tail of *lyz:dsRED* larvae shown in Fig 3 **(A, C, E, G, I)** and detailed representative merge images (brightfield and red channel) **(B, D, F, H, J)** upon their treatment with with vehicle (DMSO), 100 μM NAC (A, B), 100 μM mito-TEMPO (MT), and 100 nM tempol (T) (C, D), 250 μM apocynin (E, F), or upon genetic inhibition of *nox1* and *nox5* (G, H), and nox4 (I, J). White arrows indicate keratinocyte aggregates. Each dot represents one individual, and the mean ± SEM for each group is also shown. *p*-Values were calculated using 1-way ANOVA and Tukey multiple range test and *t* test. ****$p \leq 0.0001$. The data underlying this figure can be found in S1 Data. ANOVA, analysis of variance; NAC, N-acetylcysteine; ROS, reactive oxygen species.
(PDF)

**S3 Fig. Related to Fig 3. Efficiency of crRNA for *nox1*, *nox4*, and *nox5*.** Analysis of genome editing efficiency in larvae injected with control or *nox1* **(A)**, *nox4* **(B)**, and *nox5* **(C)** crRNA/Cas-9 complexes and quantification rate of NHEJ-mediated repair showing all INDELs (https://tide.nki.nl/). INDEL, insertion and deletion; NHEJ, nonhomologous end joining.
(PDF)

**S4 Fig. Related to Fig 4. Genetic and pharmacological inhibition of Nampt and Parp1 diminishes PARylation, skin inflammation, and restores epithelial integrity in Spint1a-deficient larvae. (A, B)** Quantification of keratinocyte aggregates and detailed representative merge images (brightfield and red channels) of wild-type and Spint1a-deficient zebrafish treated with vehicle (DMSO) or 100 μM olaparib shown in Fig 4A. **(C–E)** Neutrophil distribution (C) and keratinocyte aggregates (D) of Spint1a-deficient larvae injected with control or *parp1* crRNA/Cas-9 complexes. Representative images are shown in E. **(F)** Analysis of genome editing efficiency in larvae injected with control or *parp1* crRNA/Cas-9 complexes and quantification rate of NHEJ-mediated repair showing all INDELs (https://tide.nki.nl/). **(G, H)** Western blots with anti-PAR and anti-β-actin of tail fold (red boxed area) lysates from 3 dpf wild-type and Spint1a-deficient zebrafish larvae treated for 48 hours with 10 μM FK-866 or 100 μM olaparib. The relative abundance of PAR with respect to β-actin is shown in each lane. Each dot represents one individual, and the mean ± SEM for each group is also shown. *p*-Values were calculated using 1-way ANOVA and Tukey multiple range test and *t* test. ****$p \leq 0.0001$ (C). Representative merge images (brightfield and red channel) of *lyz:dsRED* zebrafish larvae of every group are shown (D). The data underlying this figure can be found in S1 Data. ANOVA, analysis of variance; INDEL, insertion and deletion; NHEJ, nonhomologous end joining; Nampt, nicotinamide phosphoribosyltransferase; PAR, poly(ADP-ribose).
(PDF)

**S5 Fig. Related to Figs 2 and 4. FK-866 and olaparib improve skin epithelial integrity in psoriasis mutants. (A, B)** Determination of the skin phenotype of 2.5 dpf zebrafish Atp1b1a-deficient larvae treated 1.5 days with 50 μM FK-866 or 100 μM olaparib. **(C)** Representative bright field images of zebrafish larvae of every group are shown. *p*-Values were calculated using chi-squared and Fisher exact test $^*p \leq 0.05$, $^{****}p \leq 0.0001$. The data underlying this figure can be found in S1 Data.
(PDF)

**S6 Fig. Related to Fig 5. Inhibition of parthanatos rescues morphological skin alterations of Spint1a-deficient larvae.** Quantification of keratinocyte aggregates **(A, E, H)** and detailed representative merge images (brightfield and red channels) **(B, F, I)** of wild-type and Spint1a-deficient larvae treated with vehicle (DMSO) or 10 nM NP (A, B, C, D), *aifm1* genetic inhibition (E, F) and *parga* mRNA overexpression (H, I) of zebrafish larvae shown in Fig 5. (C) Quantification of the percentage of nuclear Aifm1 positive cells (white arrows) in zebrafish skin, calculated as the ratio between the number of keratinocytes in which Aifm1 is found in the nucleus and total keratinocyte number analyzed. (D) Laser confocal microscopy Z stack of Aifm1 immunostaining (red) in 72 hpf wild-type and Spint1a-deficient larvae treated for 48 hours with 10 nM NP. Samples were counterstained with DAPI (blue). Normal keratinocytes, keratinocyte aggregates, and neuromast are shown. (G) Analysis of genome editing efficiency of larvae injected with control or *aifm1* crRNA/Cas-9 complexes and quantification rate of NHEJ-mediated repair showing all INDELs (https://tide.nki.nl/). Each dot represents one individual, and the mean ± SEM for each group is also shown. *p*-Values were calculated using 1-way ANOVA and Tukey multiple range test and *t* test. $^{***}p \leq 0.001$, $^{****}p \leq 0.0001$. The data underlying this figure can be found in S1 Data. ANOVA, analysis of variance; INDEL, insertion and deletion; NHEJ, nonhomologous end joining; NP, N-phenylmaleimide.
(PDF)

**S7 Fig. Related to Fig 6. NAD$^+$ and PAR metabolic pathways.** NAM, NMN, and NAD$^+$ can be taken up by specific transporters. NAD$^+$ biosynthetic pathways generate NAD$^+$ from different precursors, de novo pathway employs dietary Trp or alternatively quinolinic acid (QA), NAD$^+$ Salvage pathway mainly uses NAM but NMN and NR can also act as precursors. However, Preiss–Handler pathway utilizes NA. NAD$^+$ is consumed by CD38 yielding NAM and ADPR or cADPR. NNMT also reduces NAD$^+$ pool mediating the reaction between NAM and SAM to produce N-methylnicotinamide (1-MNA) and SAH. Finally, PARP1 synthesizes PAR by using NAD$^+$ as a cofactor. PAR is degraded to ADPR mediated by different PAR hydrolases which cleave specific chemical linkages (exo- or endoglycosidically). Metabolic intermediates: NFK, NAAD, and NAMN. NAD$^+$ transporter: CX43. ADPR, adenosine diphosphoribose; caDPR, cyclic ADPR; CX43, connexin 43; NA, nicotinic acid; NAAD, nicotinic acid adenine dinucleotide; NAD$^+$, nicotinamide adenine dinucleotide; NAM, nicotinamide; NAMN, nicotinic acid mononucleotide; NFK, N-formylkynurenine; NMN, nicotinamide mononucleotide; NR, nicotinamide riboside; PAR, poly(ADP-ribose); SAH, S-adenosylhomocysteine; SAM, S-adenosylmethionine; Trp, tryptophan.
(PDF)

**S8 Fig. Related to Fig 6. Differential expression profiles of genes encoding key NAD$^+$ metabolic enzymes in psoriasis.** Transcriptomic data from human psoriasis (GDS4602) samples from the GEO database. Nonlesional and lesional psoriasis skin were compared with healthy skin samples. Each dot represents one individual, and the mean ± SEM for each group is also shown. *p*-Values were calculated using 1-way ANOVA and Tukey multiple range test (A) and *t* test (B). ns, not significant. $^*p \leq 0.05$, $^{**}p \leq 0.01$, $^{***}p \leq 0.001$, $^{****}p \leq 0.0001$. The data

underlying this figure can be found in S1 Data. ANOVA, analysis of variance; GEO, Gene Expression Omnibus; NAD⁺, nicotinamide adenine dinucleotide.
(PDF)

**S9 Fig. Related to Fig 6. Differential expression profiles of genes encoding parthanatos components in psoriasis.** Transcriptomic data from human psoriasis (GDS4602) samples from the GEO database. Nonlesional and lesional psoriasis skin were compared with healthy skin samples. Each dot represents one individual, and the mean ± SEM for each group is also shown. $p$-Values were calculated using 1-way ANOVA and Tukey multiple range test (A) and $t$ test (B). ns, not significant. $^*p \leq 0.05$, $^{**}p \leq 0.01$, $^{***}p \leq 0.001$, $^{****}p \leq 0.0001$. The data underlying this figure can be found in S1 Data. ANOVA, analysis of variance; GEO, Gene Expression Omnibus.
(PDF)

**S1 Table. Compounds used in this study.**
(DOCX)

**S1 Data. Underlying data.**
(XLSX)

## Acknowledgments

We warmly thank I. Fuentes and P. Martínez for their excellent technical assistance and the staff of the Dermatology Service of the University Hospital Virgen de la Arrixaca for patient sample collection. We also thank Profs. S.A. Renshaw and P. Crosier for the zebrafish lines.

## Author Contributions

**Conceptualization:** Francisco J. Martínez-Morcillo, Joaquín Cantón-Sandoval, Ana B. Pérez-Oliva, Diana García-Moreno, Victoriano Mulero.

**Data curation:** Francisco J. Martínez-Morcillo, Joaquín Cantón-Sandoval, Francisco J. Martínez-Navarro, Teresa Martínez-Menchón, Raúl Corbalán-Vélez, Matthias Hammerschmidt, José C. García-Borrón, Alfonsa García-Ayala, María L. Cayuela, Ana B. Pérez-Oliva, Diana García-Moreno, Victoriano Mulero.

**Formal analysis:** Francisco J. Martínez-Morcillo, Joaquín Cantón-Sandoval, Francisco J. Martínez-Navarro, Idoya Martínez-Vicente, Joy Armistead, Julia Hatzold, Azucena López-Muñoz, Teresa Martínez-Menchón, Raúl Corbalán-Vélez, Matthias Hammerschmidt, José C. García-Borrón, Alfonsa García-Ayala, María L. Cayuela, Ana B. Pérez-Oliva, Diana García-Moreno, Victoriano Mulero.

**Funding acquisition:** Victoriano Mulero.

**Investigation:** Francisco J. Martínez-Morcillo, Joaquín Cantón-Sandoval, Francisco J. Martínez-Navarro, Isabel Cabas, Idoya Martínez-Vicente, Joy Armistead, Julia Hatzold, Azucena López-Muñoz, Ana B. Pérez-Oliva, Diana García-Moreno.

**Methodology:** Jesús Lacal.

**Supervision:** María L. Cayuela, Ana B. Pérez-Oliva, Diana García-Moreno, Victoriano Mulero.

**Writing – original draft:** Francisco J. Martínez-Morcillo.

**Writing – review & editing:** Victoriano Mulero.

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
