## [Editor Report · Decision Letter 0]

13 Jul 2021

Dear Dr Mulero, 

Thank you for submitting your revised manuscript entitled "NAMPT-derived NAD+ fuels PARP1 to promote skin inflammation through parthanatos" for consideration as a Research Article by PLOS Biology.

Your manuscript has now been evaluated by the PLOS Biology editorial staff as well as by an academic editor with relevant expertise and I am writing to let you know that we would like to send your submission back to the original reviewers of Review Commons.

Please re-submit your manuscript within two working days, i.e. by Jul 15 2021 11:59PM.

Kind regards,

Ines

--

Ines Alvarez-Garcia, PhD

Senior Editor

PLOS Biology

---

## [Decision Letter · Decision Letter 1]

11 Aug 2021

Dear Dr. Mulero,

Thank you very much for submitting a revised version of your manuscript "NAMPT-derived NAD+ fuels PARP1 to promote skin inflammation through parthanatos" for consideration as a Research Article at PLOS Biology. This revised version of your manuscript has been evaluated by the PLOS Biology editors, the Academic Editor and the reviewers.

Reviewer #2 still has some issues that will need to be satisfactorily addressed in order for us to move forward with the manuscript. In particular, reviewer #2 wants you to provide strong evidence that NAD+ is cell permeable, indicate clearly in figures what is green or red in the microscopy images, clarify how you interpret that FK-866 reduced H2O2 release, but it increased neutrophils infiltration, and clarify how NF-kB activity as measured. This reviewer also states that you did not address the concerns that the representative images were not consistent with the quantification data shown in the figures, some figures are not presented clearly and are not explained in an easily understandable way. Finally, this reviewer also has issues with how Olaparib treatment increased DNA damage but reduced PARylation. Please, also address the remaining concerns of reviewer #3. 

In light of the reviews (below), we will not be able to accept the current version of the manuscript, but we would welcome re-submission of a much-revised version that takes into account the reviewers' comments. We cannot make any decision about publication until we have seen the revised manuscript and your response to the reviewers' comments. Your revised manuscript is also likely to be sent for further evaluation by the reviewers.

We expect to receive your revised manuscript within 3 months. 

**IMPORTANT - SUBMITTING YOUR REVISION**

*Re-submission Checklist*

*Published Peer Review*

*PLOS Data Policy*

*Blot and Gel Data Policy*

Sincerely,

Ines Alvarez-Garcia

Senior Editor

PLOS Biology

REVIEWS:

Reviewer #1: The authors have addressed all my comments, and also the comments of the other reviewers. I believe that the manuscript is quite interesting and has potential clinical relevance. 

Reviewer #2: In this revision, although authors addressed few questiones, majority of questions were not fully addressed. Please see details in blue (after authors' response) as following:

2. Fig. 1: it is quite confusing how NAD+ increases H2O2 levels? Is NAD+ cell permeable? It is not clear if NAD+ has been really up taken by cells in the larvae. If NAD+ fuels PARP1 to promote skin inflammation, why NAM treatment increased H2O2 levels but NMN precursor failed to increase skin oxidative stress? No reasonable explanation has been provided.

This is an interesting point. We have shown that exogenous NAD+ added in the water of larvae increased larval NAD+ (please, see Fig. 2K). It has been shown that neurons can take up NAD+ through CX43 (Fig. S7), so a similar mechanism may operate in larval skin. As regards, the effect of NAM and NMN, a recent study has demonstrated that NAM supplementation increased zebrafish larval NAD+; however, NA, NMN and NR failed to boost larval NAD+ level (PMID: 32197067). These results are consistent with our data.

Authors did not provide strong evidence that NAD+ is cell permeable. In contrast, lots of previous literatures showed that NAD+ is not cell permeable (see review Covarrubias AJ et al., Nature Reviews, 2021). It is still not clear how NAD+ increased H2O2. Authors did not address this question at all.

3. Fig. 1E and 1G: it is not clear what is the green channel. Similarly, there is no clear description what is red or green in many other figures.

To help the interpretation of larval pictures, we have indicated in all figures what is analyzed in each fluorescent channel.

Authors still did not fully address the question for all figures. For example, authors only showed that "Representative merge images (green and red channel) of lyz:dsRED zebrafish larvae of every group are shown (E, G)." in the figure legend, which does not clearly tell what is red and green.

4. Fig. 1K and 1L: It is hard to understand why FK-866 reduced H2O2 release, but it increased neutrophils infiltration. How to interpret this conclusion?

Authors still did not address this question at all. This question is different from question 5.

6. Fig. 2I-J: it is not clear how NF-kB activity was measured. Is that based on green fluorescence shown in Fig. 2J? if so, the representative images were not consistent with the quantification data shown in I. Similarly, many other representative images were also not consistent with their quantification data throughout the manuscript. For example, Fig. 3C/D, 3E/F, 3G/H, 3L/M, Figure S2C/D, S2G/H, Fig. 4C/D, 4J/K.

The quantification of NFkB was measured in the skin, as it has already been reported previously (Candel et al., 2014). This is indicated in M&M section. The images show the whole larvae and NFkB is expressed at high levels in different tissues, such as neuromasts. To clarify this, we have included an additional figure to explain the ROI used for quantification of H2O2 and NfkB (Fig. S1G).

It is still unclear how NF-kB activity as measured. It was also not really indicated in M&M sections as authors mentioned. Authors also did not address the concerns that the representative images were not consistent with the quantification data shown in Fig. 3C/D, 3E/F, 3H/I, 3L/M, Figure S2C/D, S2G/H, Fig. 4C/D, 4J/K.

7. Figure S1C, Nampta/Namptb protein expression should be checked and shown after its KO using crispr/cas9 technique.

Unfortunately, we have used to different antibodies and both failed to crossreact with zebrafish Nampta/b. However, we have included the efficiency of CRISPR-Cas9 in Fig. S1F of the revised version. The efficiency is relatively low, probably indicating that is indispensable for zebrafish development, as occurs in mice (PMID 28333140).

Fig. S1F was not presented clearly and was not explained in an easily understandable way.

8. Fig. 3I: protein expression of nox1, nox4 and nox 5 should be checked after genetic inhibition using CRISPR/Cas9 technique.

Unfortunately, we do not have antibodies able to recognize zebrafish Nox1, Nox4 and Nox5. However, we have provided the efficiency of the gRNA used for each gene (Fig. S3) and it is about 65%.

Fig. S3 was not presented clearly and was not explained in an easily understandable way. It is difficulty to charge the efficiency of gRNAs based on these data. Checking their protein levels is mandatory to see if the experimental conditions are successful. 

9. Fig. 4: If Olaparib treatment increased DNA damage, will it increase PARP1 activation and PAR formation?

As it has widely used in mammalian models, parthanatos is triggered by overactivation of PARP1 following DNA damage. Therefore, although inhibition of olaparib may further induces DNA damage, it blocks parthanatos. This is consistent with our results showing that Olaparib reduces PARylation (Fig. S4H) and cell death (Figs. 4J, 4K).

No convincing evidence was provided that olaparib increased DNA damage in this manuscript. In contrast, authors showed that Olaparib reduced PARylation and cell death. PARP-1 is a DNA damage sensor and its biological functions are highly context dependent. In cancer research (without severe DNA damage), PARP-1 may contribute to DNA repair and PARP inhibitor may increase DNA damage leading to cancer cell death. However, following severe DNA damage, as authors mentioned that PARP-1 hyperactivation will promote parthanatos. It is puzzling how Olaparib treatment increased DNA damage but reduced PARylation. This conclusion sounds self-contradictory.

10. Fig. 4M: it is not clear what staining has been done. No difference was observed among different groups.

As indicated in the figure legends, pγH2Ax+ (green) keratinocytes (red) are shown. We have indicated this in the figure and include arrows to show pγH2Ax+ cells. The quantitation of this experiment (Fig. 4L) show that FK-866 robustly reduced, while olaparib increases, keratinocyte DNA damage.

The image quality of Fig. 4M is poor. Fig. 4M did not show any difference among different groups. It is quite surprising that Fig. 4N shows statistic significance among groups with such small difference. Again, if Olaparib increased DNA damage, how it reduced PARylation? PARP-1 is the DNA damage sensor and PARylation is the bioproduct of DNA damage and PARP-1 activation.

Minor

3. Figure S3 and S6E: they should be presented in an easy understandable way for the general readers.

We have explained in the legends the graph output of TIDE analysis.

Authors still did not present Fig. 1F, S3, S4F and 6F clearly and did not explain the data in an easily understandable way.

Reviewer #3: The revised manuscript has been substantially improved. However, some weaknesses that need to be addressed include:

1. The AIF pictures are very poor, better quality pictures with data needs quantification is required. In addition, cell fractionation experiments for levels of AIF in nuclear and mitochondrial fractions are critical for the interpretation of these results. 

2. While PARG mRNA has been used in the revisions but no data is provided that the PARG levels indeed increase in the cells/larvae. 

3. NAD quantification following NAMPT modulation vs PARP inhibition has not been shown. 

Overall, this is an interesting study but the above mentioned concerns need to be addressed to make the interpretations/conclusions of this study clear.

---

## [Editor Report · Decision Letter 2]

1 Oct 2021

Dear Victor,

Thank you for submitting your revised Research Article entitled "NAMPT-derived NAD+ fuels PARP1 to promote skin inflammation through parthanatos" for publication in PLOS Biology. I have now discussed your revision with the other editors and obtained advice from the Academic Editor.

Based on this advice, we will probably accept this manuscript for publication, provided you satisfactorily address the remaining data policy-related request (see below).

In addition, we would like you to consider a suggestion to improve the title:

"NAMPT-derived NAD+ fuels PARP1 to promote skin inflammation through parthanatos cell death"

We expect to receive your revised manuscript within two weeks. 

*Published Peer Review History*

*Early Version*

Sincerely,

Ines

--

Ines Alvarez-Garcia, PhD,

Senior Editor,

ialvarez-garcia@plos.org,

PLOS Biology

Fig. 1B, D, F, H, I, K, L; Fig. 2A, C, E, G, I, K, L; Fig. 3A, C, J, E, G, L; Fig. 4A, C, E, J, H, L, N; Fig. 5B, E, H, K, N; Fig. 6B, C; Fig. 7D; Fig. S1A, D, F; Fig. S2A, C, E, G, I; Fig. S3A-C; Fig. 4A, C, D, F; Fig. S5B; Fig. S6A, C, E, G, H; Fig. S8 and Fig. S9

---

## [Editor Report · Decision Letter 3]

22 Oct 2021

Dear Dr Mulero,

On behalf of my colleagues and the Academic Editor, Chaitan Khosla, I am pleased to say that we can in principle offer to publish your Research Article entitled "NAMPT-derived NAD+ fuels PARP1 to promote skin inflammation through parthanatos cell death" in PLOS Biology, provided you address any remaining formatting and reporting issues. These will be detailed in an email that will follow this letter and that you will usually receive within 2-3 business days, during which time no action is required from you. Please note that we will not be able to formally accept your manuscript and schedule it for publication until you have made the required changes.

PRESS

Sincerely, 

Ines

--

Ines Alvarez-Garcia, PhD 

Senior Editor 

PLOS Biology
